# CAN LANGUAGE AGENTS BE ALTERNATIVES TO PPO? A PRELIMINARY EMPIRICAL STUDY ON OPENAI GYM

## ABSTRACT

The formidable capacity for zero- or few-shot decision-making in language agents encourages us to pose a compelling question: *Can language agents be alternatives to PPO agents in traditional sequential decision-making tasks?* To investigate this, we first take environments collected in OpenAI Gym as our testbeds and ground them to textual environments that construct the `TextGym` simulator. This allows for straightforward and efficient comparisons between PPO agents and language agents, given the widespread adoption of OpenAI Gym. To ensure a fair and effective benchmarking, we introduce 5 levels of scenario for accurate domain-knowledge controlling and a unified RL-inspired framework for language agents. Additionally, we propose an innovative explore-exploit-guided language (EXE) agent to solve tasks within `TextGym`. Through numerical experiments and ablation studies, we extract valuable insights into the decision-making capabilities of language agents and make a preliminary evaluation on their potential to be alternatives to PPO in classical sequential decision-making problems. This paper sheds light on the performance of language agents and paves the way for future research in this exciting domain. Our code is publicly available at `https://anonymous.4open.science/r/Text-Gym-Agents-5220`.

## 1 INTRODUCTION

Proximal policy optimization (PPO) (Schulman et al., 2017) is a representative traditional reinforcement learning (RL) method, attains or exceeds human decision-making performance with a large number of environment interactions in diverse classic tasks (Jaderberg et al., 2019; Mankowitz et al., 2023), and has been treated as a popular choice for learning decision. For example, when we face a `CliffWalking` task (Brockman et al., 2016a), our first attempt may be taking PPO as a try. However PPO usually needs a large number of interactions in the environment. Conversely, Large language models (LLMs; Vaswani et al., 2017; Brown et al., 2020; Devlin et al., 2019; OpenAI, 2023) exhibit formidable zero- or few-shot decision-making capabilities based on multiple LLM calls for reasoning (Creswell et al., 2023; Wu et al., 2022; Yao et al., 2022b; Shinn et al., 2023) or manipulating internal memory (Guu et al., 2020; Park et al., 2023; Qian et al., 2023; Xu et al., 2023) and other external tools (Schick et al., 2023; Mialon et al., 2023; Qin et al., 2023). Consequently, a direct and inescapable question arises:

> *Can language agents be alternatives to PPO in sequential decision-making tasks?*

A plethora of environments have been proposed to assess the decision-making capabilities of language agents across domains, including games (Shridhar et al., 2021; Fan et al., 2022), code execution (Hendrycks et al., 2021; Zheng et al., 2023), and real-world tasks (Yao et al., 2022a; Zhou et al., 2023). Nonetheless, these environments, still in their formative stages, mainly focus on natural language understanding instead of general sequential decision-making. To this end, we adopt environments gathered by `OpenAI Gym`, the often-used sequential decision environments in RL, as our testbed. To evaluate the performance of language agents, we first ground the environments to texts with LLMs and propose a pioneering platform, `TextGym`. Due to the continuous advancements integrated into `OpenAI Gym` environments, `TextGym` not only enables a reliable and exhaustive comparison between RL and language agents in terms of performance, sample efficiency, and generalizability but also involves, in a light-weight manner, the critical challenges inherent to existing

environments specifically designed for language agents, including complex reasoning, long planning horizons, and exploration of high-dimensional policy spaces.

Nonetheless, evaluating language agents' decision-making capabilities (to decide which agents to use) in `TextGym`, which contains a set of classic sequence decision problems, with efficiency and fairness is challenging due to two factors. On the one side, varying degrees of *domain knowledge* are supplied to LLMs by different language agents while forming decisions, ranging from zero-shot prompting (Kojima et al., 2022), fine-grained few-shot (or in-context) examples (Du et al., 2023), to coarse-grained expert guidelines (Wang et al., 2023a); conversely, disparate language agents equip LLMs with distinct cognition skills, such as thinking (Wei et al., 2022), memorizing (Park et al., 2023), and reflecting (Shinn et al., 2023), with blurred boundaries in different *algorithmic frameworks*. We shall expound upon these factors, i.e., domain knowledge, and algorithmic frameworks individually.

Firstly, the design of LLM prompting for various language agents frequently incorporates disparate degrees of domain knowledge. For instance, Zero-shot-CoT (Kojima et al., 2022) merely appends *"Let's think step by step"* prior to each response, devoid of any task-related knowledge. In contrast, ELLM (Du et al., 2023) provides ground-truth examples in each round of goal generation, exemplified by *"Q: You see water, grass, cow, and diamond. You are targeting grass. You have in your inventory plant. What do you do? A: Drink water."* Furthermore, Voyager (Wang et al., 2023a) offers expert guidance for attaining elevated scores through prompting, as illustrated by *"I may sometimes need to repeat some tasks if I need to collect more resources to complete more difficult tasks. Only repeat tasks if necessary."* Intuitively, varying domain knowledge levels will unavoidably influence language agents' performance. Consequently, regulating the extent of domain knowledge is vital for ensuring fair comparisons.

Secondly, to enhance the decision-making ability of LLMs, different language agents bestow upon them various cognitive skills via the incorporation of external modules. CoT (Wei et al., 2022) introduces a series of intermediate reasoning steps in the prompt, considerably improve LLMs' capacity to conduct complex reasoning; Generative Agents (Park et al., 2023) delineate an architecture that expands an LLM to maintain the agent's experiences in natural language, synthesize memories over time, and dynamically retrieve them to plan more efficient behavior; Reflexion (Shinn et al., 2023) proposes a novel framework to endow LLMs with self-reflection capabilities, thereby facilitating planning via trial-and-error analogous to RL. However, the boundaries between different cognitive skills are blurry, with reflection potentially encompassing both thinking and memorization processes. Simultaneously, different language agents employ distinct algorithmic frameworks in pursuit of identical cognitive skills. This status quo hinders the conduction of fair comparisons and thorough ablation studies of diverse language agents. As such, it is imperative to integrate extant language agents into a unified algorithmic framework.

This paper addresses the above challenges and assesses the decision-making capabilities of language agents in a more efficient and fair manner. More concretely, we introduce a hierarchy of domain knowledge, comprising 5 levels, ranging from the absence of domain knowledge to the provision of expert strategies required for task completion. Additionally, drawing inspiration from the RL, we dissect the language agents assessed in this paper into three components: *actor*, *critic*, and *learner*, thereby assimilating these agents within a unified algorithmic architecture. Lastly, we put forth an innovative **EX**plore-**E**xploit-guided language agent (**EXE**) devised to tackle partially observable, sparse reward tasks, which pose significant challenges to all language agents examined herein. With numerical experiments and ablation studies, we summarize the following **important observations**: 1) Language agents have the potential of completing tasks in the `TextGym` for several environments (i.e., `Blackjack`, `Cartpole`, `CliffWalking`, `MoutainCar`, and `MoutainCarContinuous`) while failing at more challenging environments (i.e., `Acrobot`, `LunarLander`, and `Taxi`). 2) By engaging in environmental interactions, the performance of language agents approaches levels akin to those obtained via expert knowledge prompting. 3) Our proposed EXE achieves higher performance compared to other language agents in the `TextGym` setting.

**Remark 1:** This paper presents a preliminary study that investigates the potential of language agents in executing traditional sequential decision tasks, an area typically dominated by PPO and other RL techniques. As a nascent field of inquiry, the decision-making capabilities of language agents are still under-explored, particularly in the context of environments like `OpenAI Gym`. In this exploratory phase, we aim to unveil the competencies of LLMs in understanding and engaging with sequential

decision-making processes. Our intent is not to perform an exhaustive comparison with PPO, but rather to spark a dialogue on the adaptability and evolution of language agents within this domain. We acknowledge the limitations inherent in such an early-stage investigation, and any conclusions drawn are intended to serve as a foundation for future, more comprehensive research. This work should be interpreted as a stepping stone towards a broader understanding of the roles that language agents can play in the realm of decision-based tasks and their potential to complement or, in some instances, offer alternative solutions to conventional approaches.

## 2 BENCHMARK DESIGN

This section presents a benchmark framework that facilitates the effective and fair comparison of various language agents' decision-making capabilities in classic sequential tasks. At the same time, this framework juxtaposes their capacities against those of PPO in terms of cumulative reward thereby addressing the core question raised at this paper's outset: *Can language agents be alternatives to PPO in traditional sequential decision-making tasks?* More concretely, we elucidate upon the meticulous design process of `TextGym`, crafting a dependable validation platform for language agents and RL in Section 2.1. Subsequently, we propose a hierarchical domain knowledge framework, facilitating precise and fair comparison of language agents through accurate prompt control in Section 2.2. Ultimately, we introduce an RL-inspired unified algorithmic architecture, incorporating mainstream language agents, thus rendering penetrative ablation studies feasible in Section 2.3.

### 2.1 TEXTGYM: A FRIENDLY GYM FOR LANGUAGE AGENTS

We select `OpenAI Gym` (Brockman et al., 2016b) as our benchmark environment, owing to its extensive utilization in the assessment of PPO and other RL agents. In order to render Gym compatible with language agents, it is necessary to transform the environments into text-based representations, i.e., `TextGym`. It should be noted that we presuppose language agents possess access to the fundamental documentation of the environments. Concretely, we use the environment description, including the observation space, action space, reward function, and episode terminate conditions, for benchmarked environments in the official documentation[1] for the transformation. We construct `TextGym` by adding a translation wrapper for `OpenAI Gym`. Specifically, the wrapper wraps each observation in a mixture of "game description", "goal description", "action space description", and "observation description". To speed up the translation, we take GPT-4 (OpenAI, 2023) to make the translation, and the details are presented in Appendix C.1.

**Remark 2:** The assumption regarding the acquisition of documentation is justifiable, as documents are typically accessible in real-world applications (Harbour et al., 2001; Brockman et al., 2016a). By providing language agents with fundamental documentation, we emulate a pragmatic situation in which the agent possesses an elementary comprehension of the environment. This establishes the foundation for assessing language agents. In contrast to the prevailing handcrafted few-shot examples and heuristic prompting techniques, our assumption exhibits a higher degree of realism.

### 2.2 HIERARCHICAL CONTROL OF DOMAIN KNOWLEDGE AND SCENARIO FORMULATION

This subsection addresses the issue of inequitable comparisons among language agents due to unregulated domain knowledge in prompting. Specifically, we introduce a hierarchical organization of domain knowledge, consisting of 5 levels, and employ it to devise corresponding scenarios for regulating the domain knowledge utilized in prompting (see Appendix C.2 for all pseudo codes).

We define the first level of domain knowledge as a constraint where no external knowledge is provided, and the situation created by this constraint is termed the `Lv1:Zero Guidance` (De Peuter & Kaski, 2023; Chen et al., 2022). Achievable through a powerful, highly generalizable agent, this level represents the ultimate goal for language agents. However, as evidenced by our empirical studies and existing literature (Shinn et al., 2023), current agents are considerably far from achieving this objective. Another constraint of domain knowledge occurs when a human participant supplies valuable knowledge, expertise, or even optimal policy descriptions, which we designate as the ultimate level of domain knowledge. The situation created by this constraint is called the

---

[1]https://gymnasium.farama.org/.

`Lv5:Expert Guidance`. This approach is prevalent in contemporary language agent development. Nonetheless, human guidance is not always reliable, and the significant reliance on it poses a major challenge when deploying language agents to address general decision-making problems.

To facilitate the assessment of more practical scenarios, we propose 3 additional levels of domain knowledge and corresponding scenarios, inspired by data-driven learning paradigms. Firstly, informed by offline RL, experiences acquired through non-optimal policies can serve as a form of domain knowledge. We label such experiences as the second level of domain knowledge, and the situation created by this level is called the `Lv2:Sub-optimal Experience Guidance`. Subsequently, influenced by RL, interactions with the environment can be considered a unique form of knowledge to direct learning. We classify these interactions as the third level of domain knowledge, and the situation created by this level is termed the `Lv3:Self-Guidance` (as shown in Algorithm 1). The `Lv3` scenario enables agents to accumulate experience autonomously and implement corresponding enhancements. Finally, we identify expert or optimal policy-derived experiences as the fourth level of domain knowledge and the associated scenario as the `Lv4:Optimal Experience Guidance`, which is informed by imitation learning. These 3 scenarios explicitly control domain knowledge by imposing the same experience or number of interactions on language agents.

---

**Algorithm 1** Pseudo-code for Level 3: Self-Guidance

---

Initialize agent $M$, knowledge memory $M_k = \varnothing$.
**for** episode = 1 to N **do**
    Update agent $M$.update($M_k$)                 ▷ Update agents with the knowledge
    Collect a trajectory $\tau$ with $M$ in the environment.        ▷ Rollout
    Append $\tau$ to $M_k$.                         ▷ Update the knowledge
**end for**

---

Viewed from another angle, the complexity of scenario formulation generally increases with each level. Constructing a `Lv1` scenario is relatively simple for any task. For `Lv2` and `Lv4`, the challenges involve formulating corresponding policies and implementing them in real world or simulators to gather experiences. For `Lv3`, the challenge lies in deploying an unstable and evolving policy to collect experiences and update accordingly. For example, in autonomous driving, numerous suboptimal trajectories can be readily obtained, thus establishing `Lv2`. `Lv3` requires the deployment of agents in real-world settings or simulators, which are more difficult to develop. For `Lv4`, researchers must initially design expert policies and subsequently collect expert trajectories. The final level, `Lv5`, necessitates that researchers create knowledge explicitly to guide the agent in making safe and effective driving decisions. Although humans perform well on the road, it remains an open question how to construct human knowledge to assist language agents in accomplishing the task. This challenge becomes even more difficult for tasks that humans cannot complete.

## 2.3 RL-INSPIRED UNIFIED ALGORITHMIC FRAMEWORK FOR LANGUAGE AGENTS

In this subsection, we propose a comprehensive framework (Figure 1) for the conceptualization of language agents, drawing inspiration from RL principles. The language agents consist of three components: the *actor*, *critic*, and *learner*, where the actor is tasked with action selection based on the current environmental state, the critic is responsible for assessing the quality of actions executed by the actor, and the learner is accountable for updating the actor and critic in response to feedback obtained from the environment.

**Actor.** The majority of existing language agents predominantly emphasize actor design. Three components are involved for an actor: the profile, memory, and action instruction. The profile delineates its features, the action instructions specify the ways to generate the final action response, and the memory establishes its capacity for retaining historical information. Despite the diversity of profile designs, we classify them according to their adopted style, rather than concentrating on specific "magic words or prompts." The prompt style represents the manner in which researchers formulate prompts for the actor. Numerous prompting variants have been proposed, which can be broadly categorized into machine language, free-form verbal, and structured verbal. Machine language prompts direct the language agent to produce programs as opposed to natural language. Structured verbal prompts instruct language agents to generate output in an organized fashion (e.g., SaP; Press et al., 2022) produces output in the "*Follow up→Intermediate answer*" format), while free-form verbal

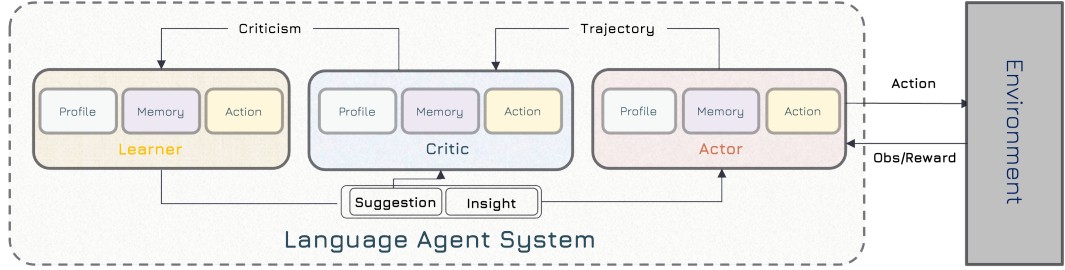

Figure 1: Illustration of the actor-critic-learner framework. A typical language agent is composed of 3 components: the actor, critic, and learner, each with its profile, memory, and action instructions. Initially, the actor interacts with the environment, collecting a trajectory. Subsequently, the critic evaluates the trajectory and provides feedback to the learner. Finally, the learner offers suggestions to the actor based on the critique, forming an iterative process of sampling and learning.

prompts impose no structural constraints. Regarding action instructions in the actor, two primary methods exist single-path and multi-path action generation. The majority of agents employ single-path generation, which directly yields the current action, while a select few utilize multi-path action generation, generating the actions in a tree-like structure. In terms of memorization, most language agents do not directly retain environmental history, whereas Reflexion (Shinn et al., 2023) and our proposed method permit agents to maintain a brief memory.

**Critic.** We classify critics in language agents into three categories: those without critics, numerical-based critics, and verbal-based critics. Numerical-based critics assess the policy by providing binary or scalar scores akin to RL critics. Capitalizing on the extensive input space acceptable to language agents, verbal-based critics offer descriptions of the policy and its performance, facilitating the conveyance of richer information.

**Learner.** As for learners, the majority of existing language agents do not incorporate learners to autonomously adjust the actor. Learners can be differentiated based on their aims, either learn to exploit or learn to explore and exploit. We provide a summary of existing language agents within the context of this framework in Table 1. We posit that this categorization enhances clarity in comparisons and fosters further development in language agent design.

Through the incorporation of language agents within the actor-critic-learner paradigm, it becomes feasible to discern the salient features of language agents. Furthermore, this paradigm facilitates the adaptation of concepts from RL in the development of language agent architectures.

**Remark 2:** It is important to note that many language agents do not directly account for the data collected in Lv2 to Lv4. To facilitate equitable comparisons, we incorporate a default critic and learner component for these agents. The critic summarizes the trajectory directly while the learner takes the summarized information as a suggestion to the actor. See Appendix C.3 for details.

## 3 EXPLORE-EXPLOIT GUIDED LANGUAGE AGENT (EXE)

Consider the cliff-walking scenario as an illustrative example: agents are initially unaware of the locations of cliffs and the goal. It is, therefore, imperative that agents engage in exploration within the environment to acquire this information. Subsequently, they must leverage such information for exploitation to develop an optimal strategy. The principle of maximum-entropy reinforcement learning (RL) (Haarnoja et al., 2018) takes into account both exploration and exploitation within its gradient propagation, demonstrating commendable outcomes in scenarios characterized by partial observability and sparse rewards. Inspired by this, we introduce the Explore-Exploit Guided Language Agent (EXE), designed to direct the actor with a dual emphasis on exploration and exploitation, as illustrated in Figure 2 and delineated in Algorithm 4.

**Actor:** The actor, a language model, consists of three primary components. Initially, it accepts the game description, state, and action space description as input, analogous to reinforcement learning configurations. Subsequently, a suggestions component receives input from the learner prior to the

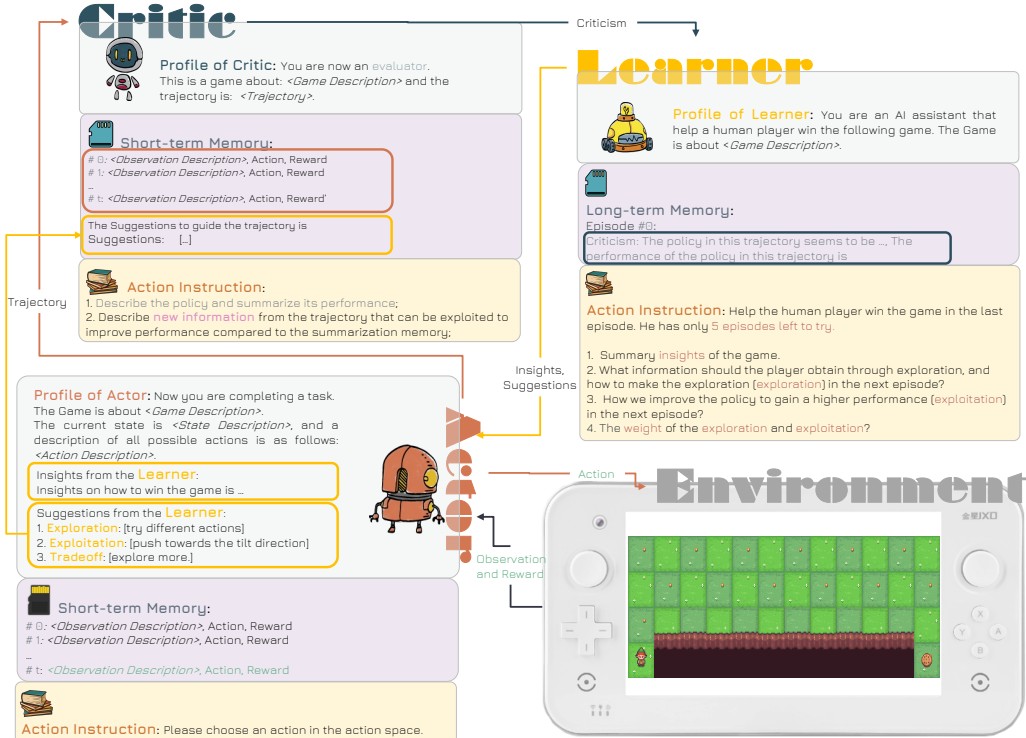

Figure 2: Illustration of `EXE` in the `CartPole-v0` game, featuring three main components. **Actor** receives insights and suggestions from the learner and interacts with the environment, **Critic** evaluates the trajectory generated by the actor based on the suggestions used by the actor. The **Learner** processes criticism from the critic and provides insights and suggestions to the actor.

commencement of each episode, corresponding to the gradient descent and exploration framework in reinforcement learning. In addition to these elements, the actor possesses a short-term memory module that retains experiences from the current episode, analogous to the recurrent design in reinforcement learning policies. Utilizing these components, the actor directly selects actions and engages with the environment.

**Critic:** The critic is characterized by a combination of memory, game description, trajectory, suggestion, and critic instruction. The game description informs the critic of its environmental context, while the trajectory provides the necessary information for evaluation. The suggestion tells what the trajectory is for and what information should be noted. Upon acquiring this information, the critic adheres to the instructions to generate criticism, which characterizes the policy and extracts novel information accordingly. This criticism is subsequently transmitted to the learner.

**Learner:** The learner receives the game description and utilizes its memory to retain the criticism provided by the critic. Prior to the initiation of each episode, the learner processes the game description and its memory as the current state, adhering to instructions that explicitly account for the number of episodes in order to offer suggestions regarding 1) exploration strategies, 2) exploitation methods, and 3) the optimal balance between exploration and exploitation in subsequent episodes. These suggestions are then conveyed to the actor. It should be noted that our `EXE` always generate suggestions even if there is no trajectory obtained (`Lv1` and `Lv5`)

## 4 EXPERIMENTS

This section performs a series of experiments to assess the efficacy of our proposed `EXE`, alternative language agents within the `TextGym`, and PPO agents within `OpenAI Gym`. Our investigation seeks to address the following research questions: **Q1:** Are language agents capable of achieving

performance levels comparable to those of PPO? **Q2:** What is the influence of domain knowledge controlling on language agent performance? **Q3:** How does the design of a language agent affect its performance?

**Environment and Scenario Configurations.** Grounded environments include *Classic Control* (`Acrobot-v1`, `CartPole-v0`, `MountainCar-v0` and `MountainCarContinuous-v0`), *Toy Text* (`Blackjack-v1`, `Taxi-v3`, `CliffWalking-v0`), and *Box2D* (`LunarLander-v2`). Each environment poses distinct challenges for agents, as outlined in Appendix C.1. Moreover, we devise the 5-level domain knowledge controlling scenario for each environment. For `Lv2`, we employ a random policy to gather 5 trajectories per environment, offering a rudimentary approach to scenario creation. For `Lv3`, we allow each agent to interact with the environment across 5 episodes. For `Lv4`, we train cutting-edge policies based on Tianshou (Weng et al., 2022) for each environment to collect 5 expert trajectories. To create `Lv5` scenarios, we meticulously design the scenarios with human input. Specifically, we allocate 1 hour of effort to develop scenarios for a single agent in each environment, randomizing the sequence of language agents. These scenarios are crafted to evaluate the agents' capacity to leverage information and perform in a variety of contexts.

**Language Agent Configurations.** For language agents lacking a critic or learner, we implement them with default one, as elaborated in Appendix C.3. This implementation does not modify their behavior in `Lv1` and `Lv5` scenarios but allows them to exploit information in `Lv2` to `Lv4` scenarios. We implement agents using OpenAI APIs[2], with all agents employing the `gpt-3.5-turbo0301` model if not specifically mentioned. For more details, we refer the reader to Appendix C.3

We conduct extensive experiments involving all agents across every scenario within the given environments with 5 seeds if not specifically mentioned. To assess performance across diverse environments, we initially establish a solvability threshold $l$ and a state-of-the-art threshold $h$ as in Table 3. The solvability threshold evaluates the extent to which the primary objective[3] of the task is achieved, while the optimal threshold is derived from RL agents employed in constructing the `Lv4` scenario. Subsequently, we normalize the performance $r$ to $r_n$ by setting $r_n = \frac{r-l}{h-l}$ if $r > l$ while $r_n = -1$ for $r <= l$. When the value surpasses 0, it signifies that agents can access the solvability zone, and values nearing 1 indicate closeness to state-of-the-art performance. We take the median performance of the agent across 5 seeds its performance when not specifically mentioned. For a more detailed description of the evaluation process, refer to Appendix C.5.

## 4.1 INVESTIGATING THE POTENTIAL OF LANGUAGE AGENTS.

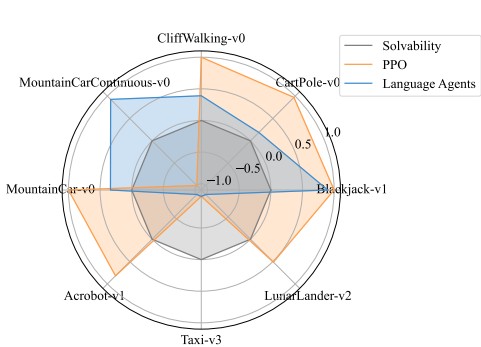

(a) Solvability results across environments.

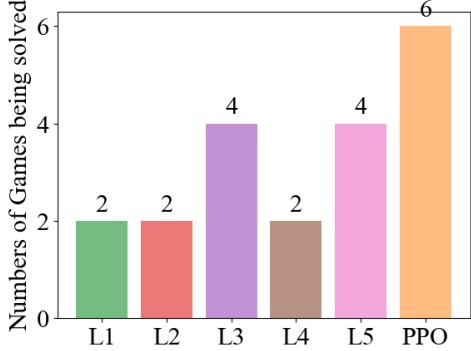

(b) Solvability results across scenarios.

Figure 3: The radar graph and histogram comparing language agents and PPO. For Figure 3a, the gray area demarcates the solvability threshold, and beyond this region, the task can be completed by the algorithm. Figure 3b displays the solvability, with PPO solving 6 out of 8 environments and language agents solving 5. This figure further examines the agent's performance in specific scenarios, showing that language agents in `Lv3` and `Lv5` solve 4 games, while others solve less.

---

[2]https://openai.com/blog/openai-api.

[3]E.g., the minimal return of reaching the goal without stepping into cliffs is set as the solvability threshold of `Cliffwalking`.

Figure 3a illustrates the median value of the normalized performance of the top-performing language agents for each environment. It is evident that language agents can successfully solve 5 out of the total 8 environments, demonstrating the potential of language agents. Furthermore, in our experimental setup, language agents require no more than 5 episodes to achieve this performance, whereas the PPO agent necessitates 20, 000 episodes. This highlights the superior task-relevant interaction efficiency of language agents in comparison to RL.

However, there are 3 environments (`LunarLander-v2`, `Taxi-v3`, and `Acrobot-v1`) where no language agents can attain the solvability threshold. This reveals that language agents still face difficulties in addressing the challenges presented by these environments. Such tasks typically involve extremely partial observations and complex and stochastic dynamics. We posit that further advanced design are needed to overcome these challenges and render the tasks manageable.

**Key Observations:** 1) Language agents exhibit good performance in environments that have simple dynamics. They can achieve the solvability threshold in these environments with significantly fewer interactions or data compared to PPO. 2) Additional efforts are required to design language agents designed to achieve similar performances to PPO in more challenging `TextGym` environments.

## 4.2 Examining the Impact of Scenario Variations

Figure 3b presents the maximal performance and median performance achieved by language agents in each environment, considering various scenario levels. It is evident that agents in `Lv3` and `Lv5` outperform their counterparts, successfully solving 4/8 environments in the median performance. Agents in `Lv1`, `Lv2`, and `Lv4` can only solve 2/8 environments indicating the zero-, sub-optimal, and optimal experience guidance are less helpful in achieving great performance in median.

**Key Findings:** 1) The `Lv3` scenario is effective and should be prioritized when designing a prompt.

## 4.3 Exploring Language Agents Choice

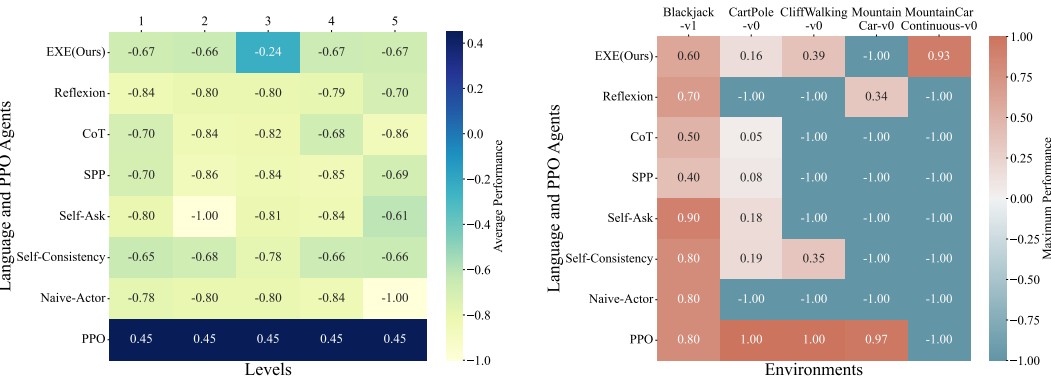

(a) Average performance across environments.  (b) Maximum performance across levels.

Figure 4: Heatmap for the performance of language agents in different scenarios and environments.

**Given a specific scenario, which language agent should be preferred?** Figure 4a displays the average performance across all solvable environments for each language agent under different scenario levels. It is evident that the proposed `EXE` significantly surpasses other agents in the `Lv3` scenario, emphasizing the importance of active exploration and exploitation in this context. In the `Lv1`, `Lv2`, `Lv3` and `Lv5` scenario, `EXE` and Self-Consistency achieve similar performance and are not outperformed by other agents significantly. Thus `EXE` is generally preferred.

**Given a specific environment, which language agent should be preferred?** We visualize the maximal performance across scenarios that each language agent can achieve for the environments in Figure 4b. `EXE` solves 4 environments while others can not solve more than 3 environments, demonstrating the superiority of our method. See Appendix D for extensive case studies.

**Key Findings:** 1) In the `Lv3` scenario, `EXE` outperforms other agents due to its active exploration and exploitation capabilities. 2) In environments that are solvable for language agents, `EXE` surpasses other agents with the ability to solve 4 out of 5 environments.

## 4.4 CASE STUDY BETWEEN EXE AND REFLEXION

`EXE` and Reflexion are designed for `Lv3`, however their performances diverge. Figure 5 and 6b visualize their learning processes. Relfxion does not learn to avoid cliff in `CliffWalking-v0` and fails to reach the goal in `MoutainCarContinuous-v0`. While `EXE` learns to improve median performances reach to the solvability threshold at the last episode for the two environments.

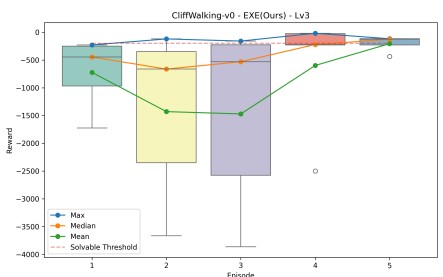
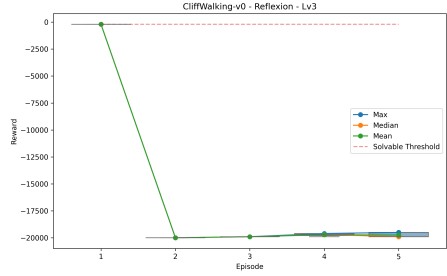

(a) The performance of EXE in `Lv3` scenario.

(b) The performance of Reflexion in `Lv3` scenario.

Figure 5: The performance of `EXE` and Reflexion in the `CliffWalking-v0` environment at `Lv3`.

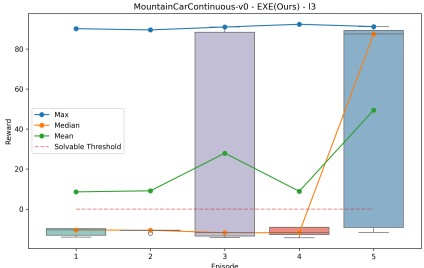
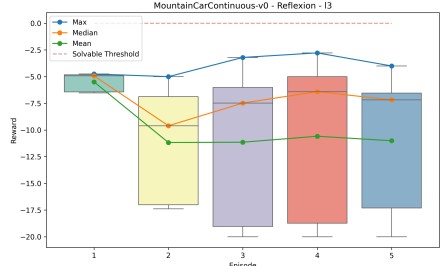

(a) The performance of EXE in `Lv3` scenario.

(b) The performance of Reflexion in `Lv3` scenario.

Figure 6: The performance of EXE and Reflexion in `MountainCarContinuous-v0` at `Lv3` with 5 successive episodes.

## 5 CONCLUSION

This paper introduces a benchmark for evaluating language agents in RL-dominated, classic decision-making problems. This benchmark encompasses a language-agent-friendly simulator, `TextGym`, a domain-knowledge-controlled hierarchical scenarios, and a unified language agents framework. We assess language agents' capabilities in approaching PPO, the influence of domain knowledge construction, and the implications of agent design. Moreover, we present a novel language agent, `EXE`, inspired by the exploration and exploitation principles inherent in RL. `EXE` demonstrates superior performance in certain scenarios compared to other language agents. We conclude by outlining many key insights regarding both language agent design and domain knowledge incorporating, which we believe will contribute to a deeper understanding of language agents and inspire further research in this domain.

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

# Supplementary Material

## Table of Contents

## A   PROBLEM FORMULATION

This paper aims to solve a set of sequential decision-making problems which each can be characterized by a partially observable Markov decision process (POMDP; Sutton & Barto, 2018) $M\langle \mathcal{S}, \mathcal{A}, \mathcal{P}, \mathcal{R}, \Omega, \mathcal{O}, \gamma \rangle$, by utilizing RL or language agents. For each problem, the environmental state is denoted by $s \in \mathcal{S}$. At each discrete timestep $t$, the agent observes an observation $o_t \in \Omega$ based on the emission or observation function $\mathcal{O}(o_t \mid s_t)$, which is a probability distribution over observations conditioned on the current state $s_t$. The agent maintains a belief state $b_t$ over the environmental states, which is updated based on the observation $o_t$ and the previous action $a_{t-1}$. The agent then opts for an action $a_t \sim \pi(a_t \mid b_t) \in \mathcal{A}$, based on the general policy $\pi : \mathcal{B} \times \mathcal{A} \to [0, 1], \forall i$ and the current belief state $b_t$. This action consequently leads to the subsequent state $s_{t+1} \in \mathcal{S}$ according to the transition function $P(s_{t+1} \mid s_t, a_t)$ and procures a scalar reward $r_t = \mathcal{R}(s_t, a_t) \in \mathbb{R}$. The principal objective for an agent is to determine a policy $\pi$ that maximizes the total expected cumulative reward, $J = \mathbb{E}_{a_t \sim \pi(\cdot|b_t), s_{t+1} \sim \mathcal{P}(\cdot|s_t, a_t)}[\sum_{t=1}^{T} \gamma^{t-1} r_t(s_t, a_t)]$, via learning from transitions during environment interaction. It is noteworthy that within the context of language agents, the state, observations, actions, and rewards are manifested through textual representations.

We aspire to benchmark the two learning paradigms, RL and language agents, within the context above, spanning the diverse tasks encompassed by TextGym and focusing on expected cumulative rewards and sample complexity. Owing to the in-context learning ability of LLMs (Brown et al., 2020), language agents naturally boast substantial superiority in the latter aspect (which shall be thoroughly validated in the experiments). More concretely, RL necessitates a multitude of magnitudes more interactions with the environment to accomplish a task from scratch, compared to human capabilities (Mnih et al., 2015; Badia et al., 2020; Vinyals et al., 2019). In contrast to RL, language agents employ language as a universal interface, substantially diminishing the need for interaction samples through their in-context learning capability and vast decision-making knowledge gleaned from copious online text.

# B  RELATED WORK

## B.1  RELATED BENCHMARKS

As large language models (LLMs) have demonstrated remarkable capabilities in generalization and planning, an array of executive benchmarks has been proposed to assess their proficiency. Initial benchmarks primarily focused on text-based games such as ALFWorld (Shridhar et al., 2021), Jericho (Hausknecht et al., 2020), and TextWorld (Côté et al., 2018).

Contemporary research endeavors have sought to appraise LLMs' performance in numerous real-world tasks beyond text games. Pioneering works like APPS (Hendrycks et al., 2021), HumanEval (Chen et al., 2021), and MBPP (Austin et al., 2021) emphasize code execution as a means to assess LLMs for functional correctness rather than text similarity. Subsequent studies (Li et al., 2022; Zheng et al., 2023; Xu et al., 2022; Nijkamp et al., 2022) have adopted this paradigm, further solidifying its prominence.

A multitude of games has been devised to investigate LLMs' task-planning aptitude. Examples include approaches utilizing the Minecraft environment (Zhu et al., 2023; Wang et al., 2023a;c) to gauge LLMs' efficacy in planning and decision-making. Tachikuma (Liang et al., 2023) employs a TRPG game log to measure LLMs' capacity to interpret and deduce intricate interactions involving multiple characters and unanticipated objects. MindAgent (Gong et al., 2023) introduces a virtual kitchen setting to evaluate LLMs' planning and coordination competencies.

Additionally, several studies have constructed interactive settings that simulate real-world scenarios. WebShop (Yao et al., 2022a) establishes a mock e-commerce website to assess LLMs' capabilities in product search and retrieval, while WebArena (Zhou et al., 2023) provides a thorough website environment encompassing an array of domains.

Diverging from the aforementioned benchmarks, our research aims to ascertain whether language agents can rival PPO in sequential decision-making tasks. Our primary focus is on decision-making capabilities, rather than proficiency in processing textual information and natural language understanding. To this end, we leverage environments curated by OpenAI Gym as our experimental platforms. These environments are typified by their comparatively lower-dimensional state and action spaces relative to conventional TextGames (Osborne et al., 2022), thereby focusing on presenting challenges in exploration, sparse reward, and stochastic dynamics crucial for decision learning, which renders them well-suited for assessing the performance of PPO. To adapt language agents to these environments, we introduce `TextGym`—a framework that transforms these environments into textual counterparts using their documentation and GPT-4. This transformation process is generally feasible for non-image-based environments. Should the language agents within `TextGym` demonstrate competitive performance against PPO in the OpenAI Gym settings, it would suggest the potential of language agents as viable alternatives to PPO.

## B.2  LANGUAGE AGENTS

| LLM Agent | Actor | | | Critic | Learner |
|---|---|---|---|---|---|
| | Prompt Profile | Action Instruction | Memory | | |
| ChatGPT | free-form | single-path action | - | - | - |
| CoT Wei et al. (2022) | free-form | single-path action | - | - | - |
| SaP Press et al. (2022) | structured-form | single-path action | - | - | - |
| ScP Wang et al. (2022) | free-form | multi-path action | - | - | - |
| SPP Wang et al. (2023b) | free-form | multi-path action | - | - | - |
| Reflexion Shinn et al. (2023) | free-form | single-path action | short memory | verbal/numerical-based | Learn to exploit |
| EXE (Ours) | free-form | single-path action | short memory | verbal-based | Learn to explore and exploit |

Table 1: Summary of language agents in actor-critic-learner framework.

In recent months, numerous studies have emerged that consider pre-trained large language models as agents for decision-making processes. Several of these works (Huang et al., 2022; Kojima et al., 2022; De Peuter & Kaski, 2023) enable language agents to perform zero-shot decision-making tasks. For instance, Kojima et al. (2022) employs a trigger phrase such as "*think step by step*" to prompt language agents.

In addition to these, a multitude of research endeavors (Wei et al., 2022; Gao et al., 2023; Press et al., 2022; Wang et al., 2022; 2023b;c; Sun et al., 2023; Shinn et al., 2023) seek to develop more intricate

prompts, allowing language agents to execute few-shot decision-making tasks. Wei et al. (2022); Gao et al. (2023); Press et al. (2022) formulate specific chain-of-thought and programming few-shot examples to guide the language model's appropriate behavior. Wang et al. (2022; 2023b) propose diverse action selection strategies to augment decision quality through multiple-time queries.

Distinct from the aforementioned studies, Wang et al. (2023c); Sun et al. (2023); Shinn et al. (2023) introduce innovative architectures to facilitate enhanced reasoning and decision-making capabilities for agents, as well as improved action execution with feedback.

## C  EXPERIMENTS DETAILS

### C.1  ENVIRONMENTS DETAILS

We summarize the environments of `TextGym`. The grounding process is subject to the criteria that minimal human intervention to make the grounding scalable and objective. To fulfill this criterion, we employ GPT-4 (OpenAI, 2023) for the grounding procedure. OpenAI Gym provides exhaustive documentation elucidating the essential concepts of each environment. Initially, we develop a code template (Appendix C.1) predicated on the `CartPole-v0` environment and introduce code review to make sure the code is objective. This template encompasses three classes: `ObsTranslator`, `GameDescriber`, and `TransitionTranslator`. The `ObsTranslator` is responsible for converting each observation into a textual description, while the `GameDescriber` offers an introduction and delineates the objectives of the game. The `TransitionTranslator` grounds the observation, action, reward, and consecutive observation in a textual format. Upon completion of the code template, we adhere to the generation process illustrated in Figure 7 to ground additional environments. For each environment, we supply GPT-4 with the pertinent documentation and instruct it to generate the translation code in `Python` format. Utilizing our manually crafted code example, GPT-4 generates the remaining environments in a consistent manner. This grounding ensures that our environments obviate the need for GPT-4 queries during sampling, substantially reducing temporal and financial expenditures. In essence, we manually craft one environment and execute a single query for each environment translation, rendering our grounding process both efficacious and economical. To ground each environment, we first devise an example translation code as follows. Then we adopt GPT-4 to verify the code does not introduce additional information compared to the original document. After that, GPT-4 is further taken to ground all other environments as Figure 7.

Listing 1: The language grounded `CartPole-v0` example.

```python
class ObsTranslator:
    def __init__(self,):
        pass

    def translate(self, state):
        cart_position, cart_velocity, pole_angle, pole_angular_velocity =
            state
        cart_direction = "right" if cart_velocity > 0 else "left"
        pole_direction = "right" if pole_angular_velocity > 0 else "left"
        res = (f"The cart is positioned at {cart_position:.3f}, with a
            velocity of {abs(cart_velocity):.2f} towards the {
            cart_direction}. "
               f"The pole is tilted at {abs(pole_angle):.2f} radians,
                   rotating at {abs(pole_angular_velocity):.2f} radians per
                   second towards the {pole_direction}.")
        return res

class GameDescriber:
    def __init__(self):
        pass

    def describe_goal(self):
        return "The goal is to keep the pole balanced upright for as long
            as possible."

    def describe_game(self):
```

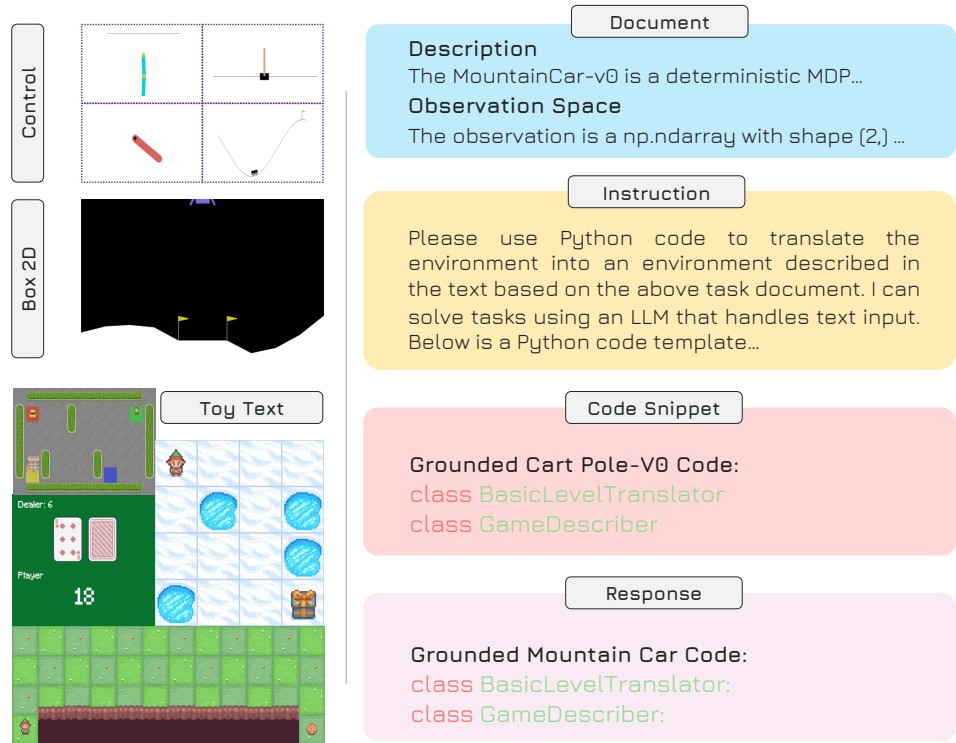

Figure 7: Illustration of grounding process. The left side of the figure presents sample target grounding environments, while the right side details the grounding process. Specifically, GPT-4 is provided with the target environment document, general grounding instructions, and a code example for grounded results in another environment. The code generated by GPT-4 is then considered the grounded output.

```
        return "In the CartPole game, you control a cart that moves along a
            horizontal track. There is a pole " \
            "standing upright on the cart. The goal of the game is to keep
                the pole balanced upright by moving the " \
            "cart left or right. The game ends if the pole tilts too far
                from the vertical position or if the cart " \
            "moves too far from the center of the track. The longer you
                can keep the pole balanced, the higher your score." \
            "Note that when the Cart Position is out of the (-2.4, 2.4)"
                zone or the Pole Angle is out of the zone (-.2095, .2095)
                "\
            ", the round ends and the game is lost. "

    def describe_action(self):
        return "Your Next Move: \n Please choose an action. Type '1' to
            push the cart to the left or '2' to push the cart to the right.
            Ensure you only provide the action number from the valid
            action list, i.e., [1, 2]."

class TransitionTranslator(ObsTranslator):
    def translate(self, infos, is_current=False):
        descriptions = []
        if is_current:
            state_desc = ObsTranslator().translate(infos[-1]['state'])
            return state_desc
        for i, info in enumerate(infos):
            assert 'state' in info, "info should contain state information"
```

```
        state_desc = ObsTranslator().translate(info['state'])
        action_desc = f"Take Action: Push {'right' if info['action'] ==
            2 else 'left'} ({info['action']})."
        reward_desc = f"Result: Reward of {info['reward']}, "
        next_state_desc = ObsTranslator().translate(info['next_state'])
        descriptions.append(f"{state_desc}.\n {action_desc} \n {
            reward_desc} \n Transit to {next_state_desc}")
    return descriptions
```

| Environment | Task | Challenge | Difficulty |
|---|---|---|---|
| `Acrobot-v1` | Swing a robot arm | Complex dynamic | Moderate |
| `Blackjack-v1` | Play Blackjack | Stochastic dynamic | Easy to moderate |
| `CartPole-v0` | Balance a pole on a cart | Inherent instability | Easy to moderate |
| `CliffWalking-v0` | Navigate in a grid world | Partial-Observe | Easy |
| `FrozenLake` | Navigate in a slippery world | Partial-Observe and stochastic dynamic | Hard |
| `LunarLander-v2` | Land a lunar module | Complex states and dynamic | Hard |
| `MountainCarContinuous-v0` | Drive a car up a hill | exploration | Moderate |
| `MountainCar-v0` | Drive a car up a hill | Sparse reward | Moderate |
| `Taxi-v3` | Pick up and drop off passengers | Complex states, partial observe and stochastic environment | Hard |

Table 2: Summary of `OpenAI Gym` environments.

## C.2 SCENARIO DETAILS

This section presents the details of each scenario. First, we propose the pseudocodes.

---
**Algorithm 2** Pseudocode for `Lv1:Zero Guidance` and `Lv5:Expert Guidance`
---
Initialize agent $M$, knowledge memory $M_k = \varnothing$ for Level 1 while $M_k = \{expert - prompts\}$.
Update agent $M$.update($M_k$)                     ▷ Update agents with the knowledge
Collect a trajectory $\tau$ with $M$ in the environment.                     ▷ Rollout

---
**Algorithm 3** Pseudocode for `Lv2:Suboptimal Guidance` and `Lv4:Optimal Guidance`
---
Initialize agent $M$, knowledge memory $M_k = \varnothing$, experiences $\tau_1, \ldots, \tau_N$.     ▷ $\tau$ is sub-optimal (optimal) trajectory
**for** episode = 1 to N **do**
    Append $\tau$ to $M_k$.                     ▷ Update the knowledge
    Update agent $M$.update($M_k$)                     ▷ Update agents with the knowledge
**end for**
Collect a trajectory $\tau$ with $M$ in the environment.                     ▷ Rollout

## C.3 DETAILS OF LANGUAGE AGENTS

We set the temperature as $0$ to reduce the uncertainty raised by the LLM. It should be noted that GPT-3.5/4 has unavoidable stochastic even set the temperature as $0$. For `EXE`, its pseudocode in `Lv3` scenario is shown in Algorithm 4.

---

**Algorithm 4** Pseudo-code for Explore-Exploit Guided Language Agent (EXE) in Lv3.

---

Initialize agent $M$ (which consists of actor $M_a$, critic $M_c$, learner $M_l$) and knowledge memory $M_k = \{game\_document\}$.
**for** episode = 1 to N **do**
  **if** $M_k = \{game\_document\}$ **then**
    $suggestion \leftarrow M_l(M_k)$                  ▷ Learning without experience
    Update actor $M_a$.update($suggestion$)       ▷ Update agents with the knowledge
    Collect a trajectory $\tau$ with $M_a$ in the environment.          ▷ Rollout
  **else**
    $suggestion, insight \leftarrow M_l(M_k)$         ▷ Learning with experience
    Update actor $M_a$.update($suggestion, insight$) ▷ Update agents with the knowledge and insight
    Collect a trajectory $\tau$ with $M_a$ in the environment.          ▷ Rollout
  **end if**
  $evaluation \leftarrow M_c(\tau, suggestion)$               ▷ Evaluation
  Append $(\tau, evaluation)$ to $M_k$.         ▷ Update Knowledge
**end for**

---

For Reflexion, its algorithm is shown below and the critic uses the cumulative reward as the evaluation results.

---

**Algorithm 5** Pseudocode for Reflexion in Lv3.

---

Initialize agent $M$ (which consists of actor $M_a$, critic $M_c$, learner $M_l$) and knowledge memory $M_k = \{game\_document\}$.
**for** episode = 1 to N **do**
  Update actor $M_a$.update($M_k$)           ▷ Update agents with the knowledge
  Collect a trajectory $\tau$ with $M_a$ in the environment.          ▷ Rollout
  $evaluation \leftarrow M_c(\tau)$                 ▷ Evaluation
  $suggestion \leftarrow M_l(M_k, \tau, evaluation)$        ▷ Learning with experience
  Append $(suggestion)$ to $M_k$.          ▷ Update Knowledge
**end for**

---

For other agents that have no critic or learner, they follow the implementation of Reflexion but with a different learner prompt:

Listing 2: Prompt of the Defual Learner

```
You will be given the history of a past experience in which you were
    placed in an environment and given a task to complete. Summarize
    your trajectory and reasoning relation between your policy and the
     obtained result. Here are two examples:
{FEW_SHOT_EXAMPLES}
{game description}, {goal description}
{traj}
Memory from past attempts:
   Trial 1: [..]
   Trial 2: [..]
Summary:
```

## C.4 DETAILS OF PPO AGENT

This section primarily elaborates on the implementation of PPO within OpenAI Gym. In our experiment, we try Tianshou (Weng et al., 2022) in almost every environment and `Taxi-v3` environment with stable-baselines3 (Raffin et al., 2021), and we employed a custom neural network architecture, The PPO architecture consists of the following components:

**Policy Network**: A sequential neural network with a linear layer (input features: observation space (`obs_space`), output features: 64, bias: enabled), followed by a hyperbolic tangent activation

function, another linear layer (input features: 64, output features: 64, bias: enabled), and another hyperbolic tangent activation function.

**Value Network**: A sequential neural network with a linear layer (input features: 2, output features: 64, bias: enabled), followed by a hyperbolic tangent activation function, another linear layer (input features: 64, output features: 64, bias: enabled), and another hyperbolic tangent activation function.

**Action Output Network**: A linear layer with input features equal to 64, output features equal to the number of actions (`action_num`), and bias enabled (`bias=True`).

**Value Output Network** : A linear layer with input features equal to 64, output features equal to 1, and bias enabled (`bias=True`).

Regarding the model size, the total number of trainable parameters in our policy model is 8, 964 when the action number is 3, and the input dim is 1. This number was calculated by summing the number of elements in each parameter tensor that requires gradient computation.

All our environments maintain consistency in the total number of training iterations, with 400 epochs and 50 trajectories sampled per epoch. We conducted a grid search for the learning rate, including $\{1e - 3, 1e - 4, 1e - 5\}$, discount factor, including $\{0.99, 0.95, 0.9\}$, weight for entropy loss, including $\{0.01, 0.05, 0.1\}$, and the number of repeat times for policy learning, including $\{10, 20\}$.

## C.5 EVALUATION DETAILS

For `Blackjack-v1` which possesses strong uncertainty, we evaluate all agents for 100 episodes at Level 3 and 20 episodes at the other levels. We utilize the consistency between the agent's actions (encompassing "hit" and "stick") in each episode and the optimal actions (Sutton & Barto, 2018) as the reward mechanism. Specifically, when the two actions align, a reward of 1 is assigned; conversely, when they are inconsistent, a reward of 0 is given. Subsequently, the cumulative reward across 20 episodes is employed as the agent's overall reward within the blackjack environment. Under the conditions of Lv3, we divide the 100 episodes into 5 groups, with each group's cumulative reward functioning as an individual episode's reward for Lv3, which is analogous to other environments. For other environments, we take 5 seeds to evaluate each agent in a scenario of the environment. For `Lv2` and `Lv4` scenarios, each agent is guided by the same 5 trajectories. For the `Lv3` scenario, each agent interacts with the environment with 5 episodes.

As for the solvability threshold, each environment has its own goal and we set the minimal cumulative reward to obtain the final goal as the threshold. For the sota threshold, we take the best RL performance as the SOTA. They are reported as Table 3.

Table 3: Solvability and SOTA thresholds for different environments.

| Metric | Blackjack-v1 | CartPole-v0 | CliffWalking-v0 | MountainCar-Continuous-v0 |
|---|---|---|---|---|
| Solvable Threshold | 10 | 40 | -200 | 0 |
| SOTA | 20 | 200 | -13 | 94.53 |
| | MountainCar-v0 | Acrobot-v1 | Taxi-v3 | LunarLander-v2 |
| Solvable Threshold | -200 | -200 | 0 | 120 |
| SOTA | -87 | -72 | 7.52 | 261 |

# D EXTENSIVE RESULTS

## D.1 ABSOLUTE PERFORMANCE COMPARISONS

See Figure 8 and Table 10.

### D.1.1 ECONOMIC AND TIME COSTS

We statistic the time and economic costs for our experiments that contribute to Table 10 in Table 4, 5 and . The time cost is caused by the frequency limit and latency in access to OpenAI API. The economic cost is caused by the service of OpenAI API. As a summary, the experiments in Table 4 consume about 64 hours and 2614 dollars.

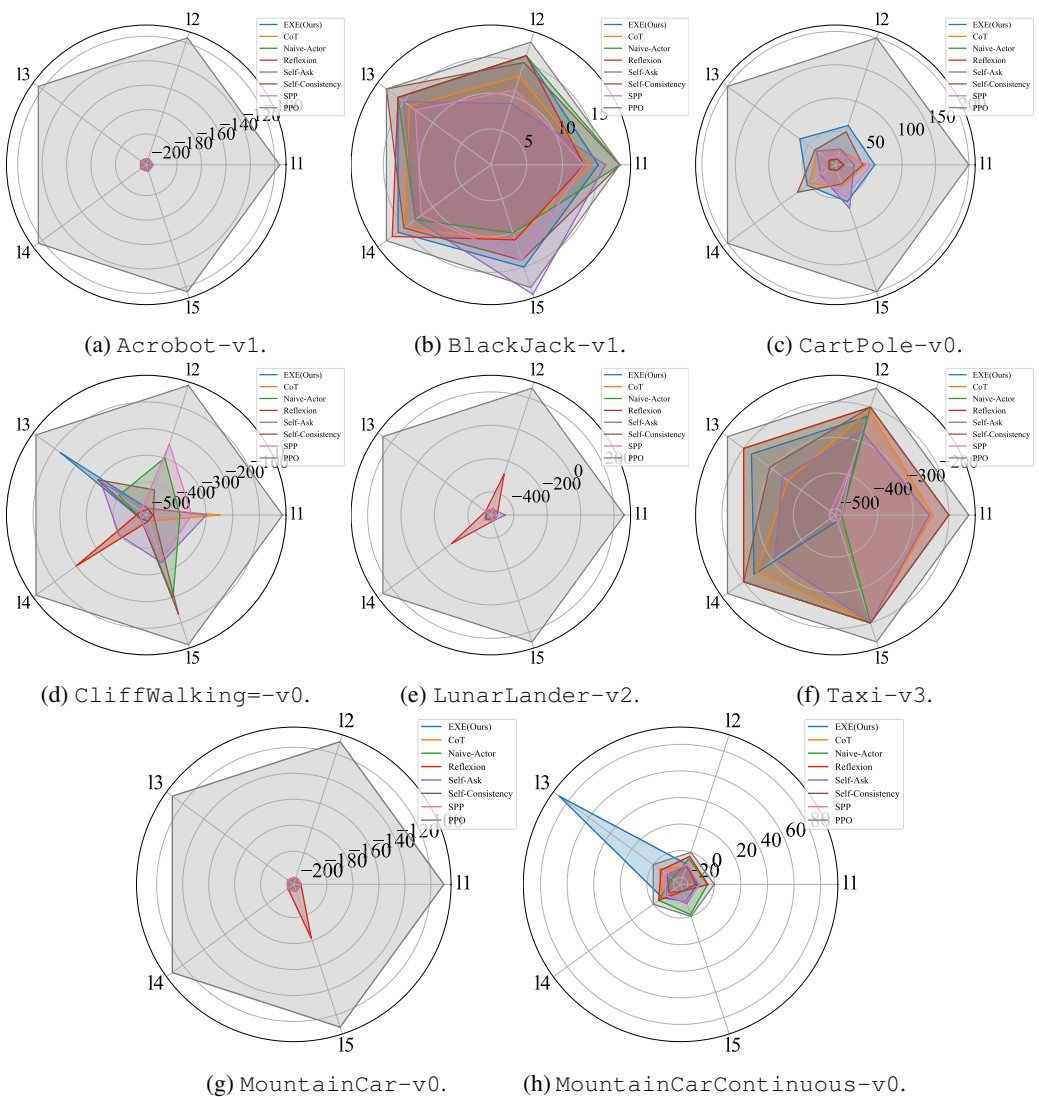

Figure 8: Radar graphs for absolute value comparison, which shows language agents' performances in different scenarios. For better representation, we clip the worst performance to $-500$.

Table 4: The total costs on time and economic for each environment.

| Game | Time Spend(s) | Economic Cost ($) |
|---|---|---|
| Acrobot-v1 | 49830 | 565 |
| Blackjack-v1 | 179 | 49 |
| CartPole-v0 | 8000 | 74 |
| CliffWalking-v0 | 23616 | 258 |
| LunarLander-v2 | 38309 | 415 |
| MountainCar-v0 | 35979 | 371 |
| MountainCarContinuous-v0 | 41571 | 390 |
| Taxi-v3 | 36072 | 489 |
| Total | 233556 | 2611 |

## D.2  EXE WITHOUT THE GAME DESCRIPTION

Here we show the performance comparison between EXE and EXE-w/o doc to verify that GPT-3.5 has a priori knowledge of the gym environment, and the results are shown in Table 6. From the table, it can be observed that in Acrobot-v1, CliffWalking-v0, LunarLander-v2, and Taxi-v3, both methods fail to achieve satisfactory results. Comparatively, in MountainCar-v0,

Table 5: The total time spent on each decider.

| Decider | Time Spend(s) |
|---|---|
| CoT | 18115 |
| EXE (Ours) | 12161 |
| Naive-Actor | 9291 |
| Reflexion | 12387 |
| SPP | 63095 |
| Self-Ask | 44640 |
| Self-Consistency | 64743 |
| Total | 224432 |

Table 6: Performance on different environments at `Lv3` with 5 successive episodes using `EXE` and `EXE` without game description (`EXE-w/o doc`). The results of the final episode are shown here. The first value in each cell is the maximum and the second is the median of different trials.

| Game | EXE | EXE-w/o doc |
|---|---|---|
| Acrobot-v1 | -200/-200 | -200/-200 |
| Blackjack-v1 | 17/16 | 19/15 |
| CartPole-v0 | 92/66 | 41/35 |
| CliffWalking-v0 | -118/-127 | -101/-794 |
| LunarLander-v2 | -656/-735 | -571/-590 |
| MountainCar-v0 | -116/-200 | -200/-200 |
| MountainCarContinuous-v0 | 91/88 | -1/-1 |
| Taxi-v3 | -200/-227 | -578/-731 |

`CartPole-v0` and `MountainCarContinuous-v0`, due to the inability of `EXE-w/o doc` to acquire pertinent knowledge about the game, its performance is markedly inferior to that of `EXE`. However, in `Blackjack-v1` setting, as GPT-3.5 possesses relevant prior knowledge, the disparity between `EXE` and `EXE-w/o doc` is not pronounced. We shall proceed to analyze the two environments of `MountainCarContinuous-v0` and `Blackjack-v1` individually.

It is important to note that in the `EXE-w/o doc`, we obscure or attenuate the environment's game description and goal description but retain the action description, allowing the knowledge of the valid action space to be preserved.

### D.2.1 CASE STUDY: MOUNTAINCARCONTINUOUS-V0

This section analyzes the result of `EXE` and `EXE-w/o doc` in `MountainCarContinuous-v0`. The prompts of the environment for `EXE` and `EXE-w/o doc` are shown below.

The prompts of the environment for `EXE` in `MountainCarContinuous-v0`

```
Now you are in the task. In the Mountain Car game, you control a car
    placed stochastically at the bottom of a sinusoidal valley. The only
    possible actions are the accelerations between -1 and 1 that can be
    applied to the car in either direction. The goal of the game is to
    strategically accelerate the car to reach the goal state on top of
    the right hill as quickly as possible. The episode ends if either the
     car reaches the goal position on top of the right hill or the length
     of the episode is 200. Your Next Move:
 Please select a numerical value within the range of [-1,1], which
     represents the directional force being applied to the car. The action
      will be limited to the range of [-1,1], and then multiplied by a
     power of 0.0015. The goal is to reach the flag placed on top of the
     right hill as quickly as possible.
```

The generated insights and suggestions of `EXE` in `MountainCarContinuous-v0`

```
The insights of the game are listed below: The key information that can
    be exploited to improve the performance of the player includes
    knowledge of the car's position and velocity, trying different
    actions and observing their effects on the car's position and
    velocity, using learned effective actions to move towards the flag as
     quickly as possible, and balancing exploration and exploitation. The
     player should also aim to improve their policy behavior by making
```

```
      more informed decisions and finding a better balance between
      exploration and exploitation. Additionally, the player should aim to
      achieve a higher final score in order to improve their overall
      performance.
The suggestions are listed below:1. The player needs to know the car's
      position and velocity to determine the best way to move the car
      towards the flag on top of the right hill. They also need to know
      which actions are effective in moving the car towards the flag.
2. The player should try different actions and observe how they affect
      the car's position and velocity. They can also try combining
      different actions to see if they work better together.
3. Once the player has learned which actions are best, they should use
      them to move the car towards the flag as quickly as possible. They
      can also try to anticipate the car's movements and adjust their
      actions accordingly.
4. The player should balance trying new actions to learn more about the
      game with using the actions they have learned are effective to
      improve their performance. They should also try to avoid taking
      actions that have not been effective in the past.
```

The prompts of the environment for EXE-w/o doc in MountainCarContinuous-v0

```
Now you are in the task.
 Your Next Move:
 Please select a numerical value within the range of [-1,1], which
      represents the directional force being applied to the car. The action
       will be limited to the range of [-1,1], and then multiplied by a
      power of 0.0015.
Your goal is to maximize the cumulative rewards for the game.
```

The generated insights and suggestions of EXE-w/o doc in MountainCarContinuous-v0

```
The insights of the game are listed below: The key information that can
      be exploited to improve the performance of the player includes
      understanding the movement of the car and how force application
      affects it, avoiding obstacles and reaching the end of the track to
      earn rewards, experimenting with different values of force and paths,
       observing other players/experts, adjusting force application and
      timing, balancing exploration and exploitation, learning from
      mistakes, and maximizing rewards through exploration and exploitation
      . The player should also try to take risks while avoiding unnecessary
       risks that could lead to failure. The policy behavior should involve
       trying different values of force, exploring different paths, and
      adjusting force application and timing to avoid obstacles and reach
      the end of the track faster. The player should also balance
      exploration and exploitation to maximize rewards and learn from their
       mistakes to improve their performance.
The suggestions are listed below:1. Understanding how the car moves and
      how force affects its movement, as well as how to avoid obstacles and
       reach the end of the track to earn rewards, is critical to determine
       the optimal policy.
2. The player should try different values of force and explore different
      paths to see how the car responds and learn from their mistakes. They
       can also observe how other players or experts play the game to gain
      insights.
3. The player can exploit the information obtained by adjusting their
      force application and timing to avoid obstacles and reach the end of
      the track faster. They can also try to earn more rewards by taking
      risks and exploring new paths.
4. The player should balance exploration and exploitation by trying new
      strategies while also sticking to what has worked in the past. They
      should not be afraid to take risks, but also not take unnecessary
      risks that could lead to failure.
```

From the aforementioned case study, it can be observed that due to the inability of the `EXE-w/o doc` to ascertain the game description and the goal description, it consequently fails to generate accurate and effective insights and suggestions. This, in turn, results in the actor being unable to make well-informed judgments.

### D.2.2    CASE STUDY: BLACKJACK-V1

This section analyzes the result of `EXE` and `EXE-w/o doc` in the environment `Blackjack-v1`. The prompts of the environment for `EXE` and `EXE-w/o doc` are shown below.

The prompts of the environment for `EXE` in `Blackjack-v1`

```
Now you are in the task.
 In the game of Blackjack, the main aim is to obtain a higher card sum
    than the dealer, but not higher than 21. Face cards count as 10,
    numerical cards hold their value, and aces can count either as 1 or
    11. The game starts with the dealer having one face-up and one face-
    down card, while the player has two face-up cards. The player can
    either choose to 'hit' (add a card) or 'stick' (stop receiving cards)
    . The game ends when the player or the dealer busts or when both the
    player and dealer are finished drawing cards.
Your Next Move: \n Please choose an action. Type '1' to stick (stop
    receiving cards) or '2' to hit (add a card). Ensure you only provide
    the action number from the valid action list, i.e., [1, 2] in json
    format.
The goal is to beat the dealer by obtaining cards that sum to closer to
    21, without going over 21.
```

The current game state and the corresponding response of `EXE` in `Blackjack-v1`

```
Current Game State:
The player's current sum is 14, the dealer is showing 10, and the player
    has a usable ace: no.

The GPT response is: Based on the current game state, the player has a
    total card value of 14, which is not very close to 21. The dealer is
    showing a 10, which is a strong card. Since the player does not have
    a usable ace, they cannot use it to improve their hand.

In this situation, the player should consider taking another card (action
     2) to try and improve their hand. However, they should be careful
    not to exceed 21 and bust.

Therefore, the optimal action for the player in this situation would be
    to take another card (action 2).
```

Case Study: The prompts of the environment for `EXE-w/o doc` in `Blackjack-v1`

```
Now you are in the task.
 Your Next Move: \n Please choose an action. Type '1' to stick (stop
    receiving cards) or '2' to hit (add a card). Ensure you only provide
    the action number from the valid action list, i.e., [1, 2] in json
    format.
Your goal is to maximize the cumulative rewards for the game.
```

Case Study: The current game state and the corresponding response of `EXE-w/o doc` in `Blackjack-v1`

```
Based on the current game state, the player's total sum is 14, and the
    dealer is showing 10. The player has two options: hit or stick.

If the player chooses to hit, they will receive another card. The new
    card could be any value from 1 to 10. If the new card is a 7 or
    higher, the player will bust and lose the game. If the new card is a
    6 or lower, the player will have a chance to improve their hand.
```

```
If the player chooses to stick, they will stop receiving cards and wait
    for the dealer to reveal their hand. The dealer will then draw cards
    until their total sum is 17 or higher. If the dealer busts, the
    player wins. If the dealer's total sum is higher than the player's,
    the player loses. If the dealer's total sum is equal to the player's,
     the game is a tie.

Based on the current game state, the player's best option is to hit. This
     is because the player's total sum is only 14, which is not a strong
    hand. By hitting, the player has a chance to improve their hand and
    get closer to 21 without busting.

Therefore, the optimal action for the player to take is to choose action
    number 2, which is to hit.

The output should be formatted as a JSON instance that conforms to the
    JSON schema below:

{"action": 2}
```

From the aforementioned case studies of `Blackjack-v1`, it is evident that GPT-3.5 possesses prior knowledge pertinent to `Blackjack-v1`. Regardless of whether users provide a game description or goal description, the `EXE` method is able to execute superior actions in response.

### D.3   EXE IN MORE ENVIRONMENTS

We have accomplished the translation of the FrozenLake-v1 environment and conducted experiments using EXE Lv3 on it, with each trial consisting of five successive episodes. The performance is illustrated in the following table 7. Evidently, EXE Lv3 is unable to resolve the `FrozenLake-v1` environment well.

Table 7:  The performance of `EXE` with 5 successive episodes using GPT-3.5 at Level 3 in the `FrozenLake-v1` environment. The first value in each cell is the median of different seeds and the value inside the parentheses represents the interquartile range while the last value is the maximum value.

|       | Episode 1 | Episode 2 | Episode 3 | Episode 4 | Episode 5 |
|-------|-----------|-----------|-----------|-----------|-----------|
| EXE   | 0(0)/0    | 0(0)/0    | 0(0)/0    | 0(0)/0    | 0(0)/0    |

We attribute the failure to the inherent randomness of the `FrozenLake-v1` environment, where the agent has a certain probability of moving in the wrong direction and falling into a hole, thereby ending the round of the game. Consequently, EXE's exploration fails to acquire any information about the target point, making it unable to provide advantageous suggestions for the agent's actions.

### D.4   MISSING CASE STUDIES

In this section, we will present the performance differences of representative language agents in various environments, stemming from domain knowledge controlling or different architectural designs. We will delve into a detailed analysis through typical cases.

#### D.4.1   BLACKJACK-V1

In this section, we analyze the performances of language agents in `Blackjack-v1`. From Table 10, agents in `Lv1` achieve great performance and their performances can hard to be improved by injecting knowledge (`Lv2` to `Lv5`). The main reason behind this is that GPT-3.5 may have enough knowledge on accomplishing this task and inappropriate knowledge may mislead it.

**Lv1 versus Lv5:**   For CoT and Naive Actor, their maximal performances downgrade more than 5. Take CoT as an example:

Case Study: The expert few-shot examples in `Lv5` for CoT in `blackjack-v1`

```
Question:
State description: Current Game State: The player's current sum is 12,
    the dealer is showing 6, and the player has a usable ace: no.
Goal description: The goal is to beat the dealer by obtaining cards that
    sum to closer to 21, without going over 21.
Action description: Your Next Move: Please choose an action. Type '1' to
    stick (stop receiving cards) or '2' to hit (add a card). Ensure you
    only provide the action number from the valid action list, i.e., [1,
    2].

Answer:
In the specific case where the player's hand is 12 and the dealer's
    visible card is 6, according to the game rules, the dealer must have
    at least 17 points in their hand to stop taking another card. If the
    dealer's card value is relatively small, the probability of the
    dealer taking another card is relatively high. If the player takes
    another card in this situation, there is a risk of exceeding 21
    points, so the optimal strategy is to stick (stop receiving cards).
So the final answer is: stick (stop receiving cards) (Action 1).

Example 2:
Question:
State description: Current Game State: The player's current sum is 17,
    the dealer is showing 10, and the player has a usable ace: no.
Goal description: The goal is to beat the dealer by obtaining cards that
    sum closer to 21, without going over 21.
Action description: Your Next Move: Please choose an action. Type '1' to
    stick (stop receiving cards) or '2' to hit (add a card). Ensure you
    only provide the action number from the valid action list, i.e., [1,
    2].

Answer:
The player's current sum of cards is 17, while the dealer is showing a
    10, and the player does not have a usable ace. The objective is to
    have a card sum closer to 21, without exceeding that value, to beat
    the dealer. Since the player's sum is 17 and there is a risk of going
     over 21 if the player hits, it might be wiser to stick with the
    current hand and not take an additional card to avoid the risk of
    busting.
So the final answer is: stick (stop receiving cards) (Action 1).
```

These examples are correct but CoT gets misled as follows:

Case Study: The decision of CoT in `Lv5` for `blackjack-v1`

```
Prompt: {expert few-shot examples}
The player's current sum is 14, the dealer is showing 1, and the player
    has a usable ace: no.
Please select an action based on the current game state and the
    information you get. You must select the appropriate action from the
    given action descriptions and cannot refrain from taking action or
    performing any prohibited actions. Here is the action description
    below:
Your Next Move: \n Please choose an action. Type '1' to stick (stop
    receiving cards) or '2' to hit (add a card). Ensure you only provide
    the action number from the valid action list, i.e., [1, 2].
...

Response: Chain-of-Thought Method:

1. The player's current sum is 14, and the dealer is showing 1.
2. According to the rules of Blackjack, the dealer must take another card
     if their visible card is less than 17.
```

```
3. If the dealer takes another card, there is a high probability that
   they will exceed 21 and bust.
4. If the player takes another card, there is a risk of exceeding 21
   points, so the optimal strategy is to stick (stop receiving cards).
5. Therefore, the optimal action for the player is to stick (stop
   receiving cards).

Output:
```
{"action": 1}
```

The optimal action should be to hit (Action 2), not to stick (Action 1). The provided response is conservative and does not favor hitting in this scenario to potentially improve the hand. Compared to conservative reasoning, CoT in $Lv1$, which has no expert guidance can correctly make decisions in such a scenario:

Case Study: The decision of CoT in Lv1 for blackjack-v1

```
The player's current sum is 14, the dealer is showing 1, and the player
   has a usable ace: no.

Chain of Thought:
- The dealer is showing a 1, which means there is a high probability that
   the dealer's next card will be a 10, resulting in a total of 21.
- The player's current sum is 14, which is a low number and far from 21.
- The player does not have a usable ace, which means the player cannot
   use it to improve their hand.
- Therefore, the best course of action is to hit (add a card) in hopes of
   getting closer to 21 without going over.

Action: 2
```

This shows the challenge of expert guidance: *correct examples may lead to unexpected performance downgrades*.

### D.4.2   CARTPOLE-V0

In the `Cartpole-v0` environment, the `EXE` algorithm outstrips competing methods in the `Lv1` scenario with respect to median performance metrics, as evidenced in Figure 9. The primary distinction of `EXE` lies in its unique paradigm of *learning without prior experience.* This approach presents initial heuristic suggestions to the agent, which are devoid of extensive historical data. Such preliminary guidance, although rudimentary, does not necessitate a foundational dataset. Therefore, the observed enhancement in performance with `EXE` under the `Lv1` scenario could be primarily driven by the generative reasoning step it induces for the agent.

Case Study: The suggestion in prompt of EXE in Lv1 for Cartpole-v0

```
The suggestions are listed below: 1. The player needs to understand that
   the goal of the game is to keep the pole balanced upright by moving
   the cart left or right. They should also know that the game ends if
   the pole tilts too far from the vertical position or if the cart
   moves too far from the center of the track. It's important to pay
   attention to the position of the cart and the angle of the pole to
   make the right decisions.

2. The player should try different actions and observe the results to see
    what works best. They can also try to keep track of their score and
   see how it changes with different actions.

3. Once the player has gathered enough information, they can use it to
   make better decisions in subsequent episodes. For example, if they
   notice that moving the cart to the left tends to keep the pole
   balanced for longer, they can try to do that more often.
```

```
4. Balancing exploration and exploitation is important to improve
   performance. The player should try different actions to explore the
   environment and gather information, but also use the information they
    have already obtained to exploit what they know works well. It's
   important to find a balance between trying new things and sticking
   with what has worked in the past.
```

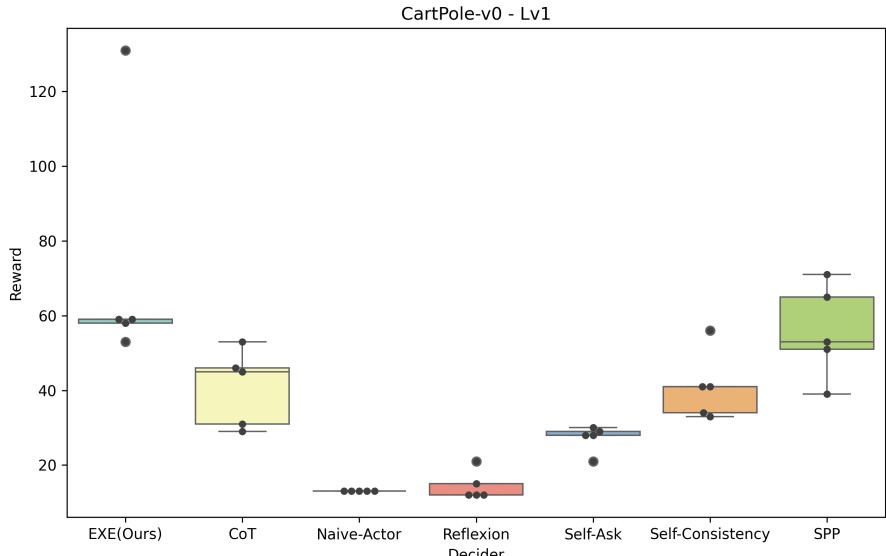

Figure 9: The performance of different deciders in `CartPole-v0` game at level 1

### D.4.3 CLIFFWALKING-V0

**A comparison of the performance between `EXE` and Reflexion at `Lv3`** We conducted 5 consecutive experiments in the `CliffWalking-v0` environment for `EXE` and Reflexion in the `Lv3` scenario, and the performance is shown in Figure 5. The pink dashed line represents the *solvable threshold* ($-200$), the blue solid line denotes the maximum value line, the orange solid line signifies the median line and the green solid line corresponds to the mean line. As can be observed, with the increase of episodes, both the median and mean of the `EXE` show growth. In contrast, the performance of Reflexion does not exhibit any improvement as the episodes progress.

Table 8: The number of goal-visiting of `EXE` and Reflexion at `Lv3` with 5 successive episodes using GPT-3.5. The first value in each cell is the number of goal visits and the second is the number of trials.

|  | Episode 1 | Episode 2 | Episode 3 | Episode 4 | Episode 5 |
|---|---|---|---|---|---|
| EXE | 5/5 | 4/5 | 3/5 | 5/5 | 5/5 |
| Reflexinon | 0/5 | 0/5 | 0/5 | 0/5 | 0/5 |

Additionally, we have also compiled the number of goal-visiting of `EXE` and Reflexion at `Lv3` with 5 successive episodes. From Table 8, it can be discerned that under the GPT-3.5, `EXE` excels at exploration, thus effectively locating the target points, while Reflexion's exploration efficiency is considerably less impressive. The case studies of `EXE` and Reflexion are shown below.

Case Study: The previous suggestion, summary, insight and current suggestion generated by GPT-3.5 using `EXE` at `Lv3`

```
The suggestion response is: 1. Critical knowledge for determining the
    optimal policy includes the layout of the grid world, the location of
     cliffs and the goal, safe actions and actions leading to cliffs or
    boundaries, and the penalties associated with each action.
2. The player should try different actions and observe the rewards
    received to explore the grid world. They can also try new actions to
    acquire information while choosing actions that have been successful
    in the past to achieve higher performance.
3. The player can exploit the information obtained by choosing actions
    that have been successful in the past and avoiding actions that lead
    to cliffs or boundaries. They can also use the optimal policy to
    navigate towards the goal while avoiding cliffs.
4. The player should balance exploration and exploitation by trying new
    actions to acquire information while also choosing actions that have
    been successful in the past to achieve higher performance. They
    should also use the optimal policy to navigate towards the goal while
     avoiding cliffs.

The summary is: 1. {
  "grid_size": "4x12",
  "cliff_locations": [
    {"x": 3, "y": 2},
    {"x": 3, "y": 1},
    {"x": 2, "y": 1},
    {"x": 1, "y": 3},
    {"x": 1, "y": 4},
    {"x": 2, "y": 4}
  ],
  "goal_location": {"x": 0, "y": 11},
  "safe_actions": [1, 2, 3, 4],
  "penalties": {
    "cliff": -100,
    "boundary": -1,
    "safe": -1
  }
}
2. The policy behavior is a combination of exploration and exploitation.
    The player tries different actions to explore the grid world and
    acquire information, while also choosing actions that have been
    successful in the past to achieve higher performance. The player also
     uses the optimal policy to navigate towards the goal while avoiding
    cliffs. However, the player's performance is poor, as they do not
    reach the goal within 200 steps and receive a performance score of
    -3863.

The insight response is: 1. Layout of the grid world, including the
    location of cliffs and the goal.
2. Safe actions and actions leading to cliffs or boundaries.
3. Optimal policy for navigating towards the goal while avoiding cliffs.
4. Balancing exploration and exploitation to acquire information and
    achieve higher performance.
5. Penalties for cliffs, boundaries, and safe actions.
6. Total reward received and performance score.
To improve performance, the player should focus on learning the optimal
    policy and safe actions, while avoiding cliffs and boundaries. They
    should also balance exploration and exploitation to acquire
    information and achieve higher performance. Additionally, they should
     aim to receive a higher total reward and performance score.

The suggestion response is: 1. Critical knowledge for determining the
    optimal policy includes the layout of the grid world, the location of
     cliffs and the goal, safe actions and actions leading to cliffs or
    boundaries, and the penalties associated with each action.
```

```
2. The player should try different actions to explore the grid world and
   observe the rewards received. They can also use trial and error to
   find safe paths to the goal while avoiding cliffs.
3. The player can exploit the information obtained by choosing actions
   that have been successful in the past and using the optimal policy to
   navigate towards the goal while avoiding cliffs.
4. The player should balance exploration and exploitation by trying new
   actions to acquire information while also choosing actions that have
   been successful in the past to achieve higher performance. They
   should also use the optimal policy to navigate towards the goal while
   avoiding cliffs.
```

Case Study: The plan generated by GPT-3.5 using Reflexion at Lv3

```
In this environment, my plan was to move right to avoid the cliff.
   However, I was stuck in a loop in which I continually tried to move
   down instead of moving right. In the next trial, I will move right if
   I am stuck in a loop again.
```

It is evident that compared to Reflexion, `EXE` extracts a greater amount of information, offering more detailed guidance.

**GPT-**4 **VS GPT-**3.5   Although `EXE` is capable of extracting some useful information, the limitations in the performance of GPT-3.5 cause it to be prone to hallucinations and result in incomplete information extraction. Consequently, the insights and suggestions provided by `EXE` may be somewhat misleading, leading to a certain degree of deviation in the agent's action selection and hindering it from reaching optimal performance. Therefore, we employ GPT-4 to evaluate its performance in the `CliffWalking-v0` environment.

We executed 5 iterations each for `EXE` and Reflexion within the GPT-3.5, allocating 200 steps per iteration. In contrast, under the GPT-4 environment, merely 2 iterations were conducted, as the decision-making component effectively resolved the problem within the second iteration. Additionally, we capped the STEP parameter at 50 steps, owing to the elevated computational cost associated with GPT-4. The performance and the cost per step are shown in Table 9. As we can see, using GPT-4 can greatly improve the performance of `EXE` and Reflexion.

We compiled the occurrences of `EXE` and Reflexion visiting the goal position during the first round of exploration under different versions of GPT. As can be seen, regardless of whether it is GPT-3.5 or GPT-4, **EXE** can locate the goal position during the initial exploration. In contrast, Reflexion does not perform as effectively when navigating to the goal position under the GPT-3.5 version. It is only with the GPT-4 version that Reflexion exhibits a 50% probability of discovering the goal position (see the second row of Table 9).

Table 9: The performance of `EXE` and Reflexion in the `CliffWalking-v0` environment when using GPT-4 and GPT-3.5-turbo at `Lv3`. The first value in each cell is the maximum and the second is the median of different trials.

|  | EXE-GPT3.5 | Reflexion-GPT3.5 | EXE-GPT4 | Reflexion-GPT4 |
|---|---|---|---|---|
| Performance | -118/-127 | -19505/-19901 | -13/-13 | -13/-50 |
| Goal found times | 5/5 | 0/5 | 4/4 | 2/4 |
| Cost(Per Step) | 0.003 | 0.008 | 0.06 | 0.05 |

Case Study: The previous suggestion, summary, insight and current suggestion generated by GPT-4 using `EXE` at `Lv3`

```
The suggestion response is: 1. Important knowledge:
  - Starting position
  - Goal position
  - Cliff locations
  - Safe paths
2. Exploration in the next episode:
```

```
     - Start by moving carefully, one step at a time
     - Try different directions to find safe paths
     - Remember cliff locations to avoid them in the future
3. Exploit information:
     - Use the safe paths found in the previous episode
     - Avoid cliffs and dangerous areas
     - Move towards the goal using the shortest path
4. Balance exploration and exploitation:
     - Begin by exploring new paths to find the shortest route
     - Once a safe path is found, exploit it to reach the goal quickly
     - If stuck, explore a new direction but remember to avoid cliffs.

The summary response is: 1. {"Starting position": "(3, 0)", "Goal
    position": "(3, 11)", "Cliff locations": ["(3, 1)"], "Safe paths":
    ["(2, 0)", "(2, 1)", "(2, 2)", "(2, 3)", "(2, 4)", "(2, 5)", "(2, 6)
    ", "(2, 7)", "(2, 8)", "(2, 9)", "(2, 10)", "(2, 11)", "(1, 11)",
    "(0, 11)", "(0, 10)", "(0, 9)", "(0, 8)", "(0, 7)", "(0, 6)", "(0, 5)
    ", "(0, 4)", "(0, 3)", "(0, 2)", "(0, 1)", "(1, 1)", "(1, 2)", "(1,
    3)", "(1, 4)", "(1, 5)", "(1, 6)", "(1, 7)", "(1, 8)", "(1, 9)", "(1,
     10)"]}
2. The policy behavior involves exploring the grid world.

The insights of the game are listed below: To help the six-year-old
    player improve their performance in the game, we can focus on the
    following key information:
1. Starting position: The player starts at position (3, 0).
2. Goal position: The player needs to reach position (3, 11).
3. Cliff locations: There is a cliff at position (3, 1) that the player
    should avoid.
4. Safe paths: There are multiple safe paths available for the player to
    reach the goal, such as moving through positions (2, 0) to (2, 11) or
     (1, 1) to (1, 10).
5. Policy behavior: The player should explore the grid world to find the
    best path to the goal.
By understanding these key points, the player can focus on navigating
    through the safe paths and avoiding the cliff while exploring the
    grid world to reach the goal position.

The suggestions are listed below:1. Important knowledge:
  - Remember the starting position (3, 0) and the goal position (3, 11).
  - Be aware of the cliff location (3, 1) and avoid it.
  - Follow the safe paths provided.
2. Exploration in the next episode:
  - Start by moving up (action 1) to avoid the cliff.
  - Then, move right (action 2) and follow the safe path.
  - Keep exploring the safe paths and avoid cliffs.
3. Exploit the information:
  - Use the safe paths to reach the goal quickly.
  - Remember the steps taken in the previous episode to avoid mistakes.
4. Balance exploration and exploitation:
  - Focus on following the safe paths and avoid unnecessary exploration.
  - Learn from previous mistakes and make better decisions in the next
      episode.
```

Through the above case study, we can observe that under the GPT-4 version, EXE is capable of extracting more abundant and useful information without experiencing hallucinations. Simultaneously, the insights and suggestions provided by EXE in the GPT-4 are more specific and effective, allowing the agent to reach the goal position more rapidly.

Case Study: The plan generated by GPT-4 using Reflexion at Lv3

```
In the previous attempt, I made a mistake by stepping into the cliff at
    location (3, 1) at the beginning, which resulted in a large penalty
    of -100. To avoid this mistake, I should move up first instead of
    moving right. My new plan is as follows:
```

```
1. Start at location (3, 0).
2. Move up to location (2, 0).
3. Move right along row 2, avoiding the cliffs, until reaching location
   (2, 11).
4. Move down to location (3, 11), which is the goal location.

By following this plan, I should be able to reach the goal location with
   a better performance score.
```

Similarly, we can see that under GPT-3.5, Reflexion does not generate useful plans, whereas under GPT-4, Reflexion generates detailed and correct plans that tell the actor the optimal moves for each step, thus allowing the actor to reach the goal point faster and gain points.

Nevertheless, during the examination of the GPT-4 version's reflexion, we discerned that the plan generated by the Reflexion is not entirely accurate, thus precluding the actor from obtaining substantial rewards while implementing the plan. We attribute this to an inadequate level of exploration in the early stages of the Reflexion, thereby hindering the discovery of valuable information. The incorrect plan generated by GPT-4 using Reflexion is shown below.

Case Study: The incorrect plan generated by GPT-4 using Reflexion at Lv3

```
In the previous attempt, I moved along the top row of the grid world,
   then backtracked and moved along the second row. This approach was
   inefficient and did not lead to the goal. In the next trial, I will
   try a more direct path to the goal by moving right and up, avoiding
   cliffs, and adjusting my path as needed.

Plan:
1. Start at location (3, 0).
2. Move right until reaching the last column (column 11).
3. Move up until reaching the top row (row 0).
4. If the goal is not found, adjust the path by moving left or down to
   explore other areas of the grid world while avoiding cliffs.
5. Continue adjusting the path until the goal is reached or the maximum
   number of steps (50) is reached.
```

**Lv4 and Lv5 outperform other levels in `CliffWalking-v0`** From Table 10, with certain deciders, agents can attain higher rewards under conditions Lv4 and Lv5 than other levels. The primary reason is that both Lv4 and Lv5 contribute to knowledge leakage to a certain degree. For example, in Lv4, the expert trajectories we provide allow the agent to deduce valuable information such as the goal position. On the other hand, in Lv5, our action responses within the human examples make mention of useful data like the goal position and cliff position, thereby enabling the agent to capitalize on this information for decision-making purposes. Here, We opt for 2 case studies featuring Lv4 from EXE and Lv5 from CoT.

Case Study: The insight and the suggestion generated by using EXE at Lv4

```
The insight response is: 1. The grid size is 4x12.
2. The start position is at [3,0] and the goal position is at [3,11].
3. There are no cliff positions in the grid.
4. The optimal policy behavior is to move right repeatedly until reaching
    the goal position, while avoiding cliffs and receiving small
   penalties for each action taken.
5. The player's current policy behavior is to repeatedly move right until
    reaching the edge of the grid, then move down one row and repeat.
6. The player incurs a total penalty of -13.0 in all episodes.
7. In episode #3, the player's policy behavior is slightly improved, but
   still suboptimal.
8. The maximum number of actions allowed is 200.

The suggestion response is: 1. The player needs to know the location of
   the cliffs and the goal, as well as the penalties for each action. It
```

```
        is also important to understand that moving towards the edge of the
     grid may not always be the best strategy.
2. The player can explore the environment by trying different actions and
     observing the resulting penalties. They can also try moving in
     different directions to see if there are any hidden paths that lead
     to the goal.
3. Once the player has obtained information about the environment, they
     can use it to make better decisions. For example, they can avoid the
     cliffs and take the shortest path to the goal.
4. Exploration and exploitation should be balanced by trying new actions
     to gather information while also using the information already
     obtained to make better decisions. The player should also be willing
     to take risks and try new strategies to improve their performance..
```

Case Study: The response of CoT in CliffWalking-v0 at Lv5

```
State description:
Current Game State: The player is at location [2, 11] in the grid world.
Goal description: The goal is to navigate from the starting point to an
    target which locate at (3,11), avoiding the cliff, in as few steps as
     possible.
Action description: Your Next Move:
Please choose an action. Type '1' to move up, '2' to move right, '3' to
    move down, or '4' to move left. Ensure you only provide the action
    number from the valid action list, i.e., [1, 2, 3, 4].

Observation:
The player is at location (2, 11) in the grid world.

Chain of Thought:
The player is at the rightmost column of the grid and the goal is also at
     the rightmost column. Therefore, the optimal action should be to
    move down, i.e., action 3.

Action:
{"action": 3}.
```

From the suggestions and insights generated by EXE in Lv4, as well as the responses from CoT in Lv5, it is evident that the agent can obtain global information pertaining to CliffWalking-v0, which in turn facilitates superior action selection.

**Self-Consistency Lv5 outperforms other deciders at Level 5 in the CliffWalking-v0 environment** From Table 10, we can observe a fascinating phenomenon: under the condition of Lv5, both the median and maximum values of Self-Consistency surpass those of other deciders. This demonstrates that, in the CliffWalking-v0 environment, Self-Consistency can not only achieve superior performance but also maintain excellence across multiple runs. The performance of different deciders at Lv5 in the CliffWalking-v0 environment is depicted in Figure 10.

We attribute its outstanding performance to two aspects: on the one hand, Self-Consistency inherits the way of thinking from CoT and the expert information of CoT at Lv5; on the other hand, Self-Consistency employs a method that selects the action with the highest frequency after multiple queries, thus reducing action outliers and making the choice of actions more deliberate and well-considered. Here comes the case study with five responses to one query with Self-Consistency.

Case Study: The responses generated by using Self-Consistency at Lv5 in the CliffWalking-v0 environment

```
Current Game State: The player is at location [3, 0] in the grid world.

Response 1:
Logic:
The player is currently at location (3, 0) in the grid world. Moving
    right is not a valid action as there is a cliff at (3, 1...10).
```

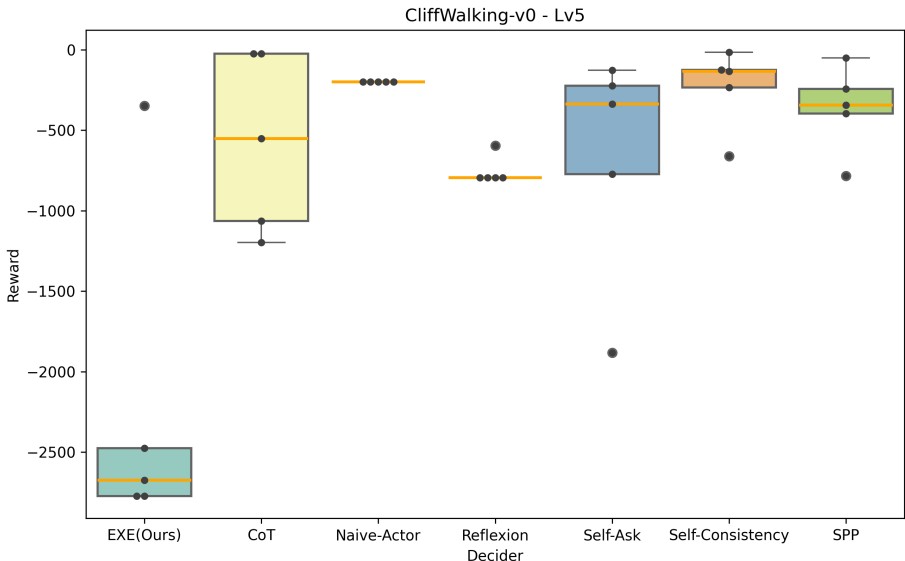

Figure 10: The performance of different deciders in `CliffWalking-v0` environment at `Lv5`. In the graph, the orange line represents the median value.

```
    Moving down is the optimal action as the target is located at (3, 11)
    , and there are no cliffs in the player's neighborhood.
Output:
{"action": 3}

Response 2:
Thought process:
Since we are trying to reach the goal in as few steps as possible, we
    need to choose the action that moves us closer to the goal. The goal
    is at position (3, 11) which is to the player's right side. Therefore
    , we should choose action 2, i.e., move right.
Output:
{"action": 2}

Response 3:
Logic:
Since the player is at the leftmost column and there is a wall to their
    left, the optimal action would be to move up (action 1). This will
    keep the player away from the cliff and still allow them to make
    progress towards the target.
Output:
{"action": 1}

Response 4:
Logic:
Since the player is at the leftmost position, the only valid actions are
    moving up or down. Moving up will take the player closer to the
    target, while moving down will take the player away from the target.
    Therefore, the optimal action is to move up, i.e., action 1.
Output:
{"action": 1}

Response 5:
Thought process: Since the target is at (3,11), it is clear that the
    player needs to move towards the right side of the grid. However,
```

```
     moving right from the current position (3,0) will lead the player to
     fall off the cliff, which will incur a large penalty of -100.
     Therefore, the player needs to move up towards row 2 to avoid the
     cliff.
Answer:
{"action": 1}.

The action list is: [3, 2, 1, 1, 1]
The optimal action is: 1.
```

From the aforementioned response on self-consistency, it is evident that precisely due to the employment of multiple samples to select the action with the highest occurrence frequency, the agent is capacitated to execute the accurate course of action when presented with a given state.

### D.4.4   MOUNTAINCARCONTINUOUS-V0

In this section, we analyze the performances of different language agents in MountainCarContinuous-v0. We conducted 5 consecutive experiments in the MountainCarContinuous-v0 environment using EXE and Reflexion at Lv3, and the resulting performance box plot is shown in Figure 6. It is evident that with the increase in episodes, both the median and mean of the EXE method are growing, while the maximum value remains at a relatively high figure. In contrast, the performance of Reflexion does not exhibit any improvement as the episodes progress. The generated suggestions from EXE are presented below.

Case Study: The suggestion generated by GPT-3.5 using EXE in MountainCarContinuous-v0 environment at Lv3

```
The suggestion response is: 1. The player needs to understand how the car
    's position and velocity change when they apply different amounts of
    force. They should also try to figure out which direction to
    accelerate in to move the car towards the goal faster.
2. The player should try different amounts of force in different
    directions to see how the car responds. They can also try
    accelerating in the same direction multiple times to see if it leads
    to faster movement.
3. Once the player has gathered information about how the car responds to
     different forces, they can use that information to make more
    informed decisions about which actions to take. For example, if they
    know that accelerating in a certain direction leads to faster
    movement, they can exploit that knowledge by continuing to accelerate
     in that direction.
4. The player should balance exploration and exploitation by trying new
    actions to gather information while also using the information they
    have already gathered to make more informed decisions. They should
    try to find a balance between trying new things and using what they
    already know to improve their performance.
```

The generated memory from Reflexion are presented below.

Case Study: The memory generated by GPT-3.5 using Reflexion in MountainCarContinuous-v0 environment at Lv3

```
Trial #0: I will try to accelerate the car to the right hill as quickly
    as possible. I will try to accelerate the car to the right hill as
    quickly as possible.
Trial #1: I will try to accelerate the car to the right hill as quickly
    as possible.
Trial #2: I have been trying to accelerate the car to the right hill as
    quickly as possible. However, I have not been able to reach the goal
    state. I should have tried to accelerate the car to the right hill as
     quickly as possible while avoiding the left hill. I will try to
    accelerate the car to the right hill as quickly as possible while
    avoiding the left hill in the next trial.
```

From the two case studies presented above, it is evident that `EXE` possesses the capacity to generate considerably effective exploration strategies based on historical trajectories, such as "*They can also try accelerating in the same direction multiple times to see if it leads to faster movement.*" Conversely, Reflexion is unable to generate accurate and useful strategies predicated on the historical trajectories found in lengthy texts, which ultimately results in the actor's inability to make optimal decisions drawing from historical information.

Simultaneously, we observed that the performance of EXE Lv4 is significantly inferior to that of EXE Lv3 in the MountainCarContinuous-v0 environment. We analyzed the suggestions provided by EXE Lv4 which are shown below. It can be noted that the suggestions generated by EXE Lv4 are excessively broad and fail to effectively guide the agent's action selection. We attribute the aforementioned issues to the following factors: On the one hand, due to the limited capabilities of GPT-3.5, it is unable to adequately summarize valuable information and conclusions from expert trajectories or mimic expert actions for making superior choices. On the other hand, during the generation of suggestions in EXE Lv3, various effective strategies are explored, as demonstrated in the aforementioned case study, which can offer more effective guidance for the agent's action selection.

Case Study: The suggestion generated by GPT-3.5 using `EXE` in `MountainCarContinuous-v0` environment at `Lv4`

```
The suggestions are listed below:1. The player needs to understand that
    the goal of the game is to reach the flag on top of the right hill as
     quickly as possible. They also need to know that the only actions
    they can take are to accelerate the car in either direction between
    -1 and 1.
2. The player should try different acceleration values to see how they
    affect the car's movement. They can also observe how the car moves in
     response to different actions.
3. Once the player has found a good acceleration strategy, they should
    stick to it and try to optimize it. They can also try to anticipate
    the car's movements and adjust their actions accordingly.
4. The player should balance exploration and exploitation by trying new
    actions to see if they improve performance, but also sticking to what
     has worked well in the past. They should also pay attention to the
    car's movements and adjust their actions accordingly.
```

### D.4.5   MOUNTAINCAR-V0

From Table 10, agents with Reflexion at `Lv5` achieve great performance than any other deciders and levels. So in this section, we analyze the performances of Reflexion in `MountainCar-v0` at `Lv5`.

We surmise that the primary reason behind this is that at `Lv5`, we have provided Reflexion with past cumulative experiences and human-generated actions along with their thought processes. However, other deciders (such as CoT, SPP, and `EXE`) and levels have not furnished corresponding information, which has led to Reflexion `Lv5`'s significantly superior performance in the `MountainCar-v0` environment when compared to other deciders and levels. The prompt for Reflexion `Lv5` is illustrated below.

Case Study: The human few-shot example of Reflexion in `MountainCar-v0` environment at `Lv5`

```
Question:
State description: Current Game State: The car is positioned at 0.472,
    with a velocity of 0.049 towards the right.
Goal description: The goal is to reach the flag placed on top of the
    right hill as quickly as possible.
Action description: Your Next Move:Please choose an action. Type '1' to
    accelerate to the left, '2' to not accelerate, or '3' to accelerate
    to the right.Ensure you only provide the action number from the valid
     action list, i.e., [1, 2, 3].
Your memory for the task below:
Trial 0:
In this environment, my plan was to accelerate the car to the right as
    quickly as possible. However, the goal of the game is to reach the
```

```
      flag placed on top of the right hill as quickly as possible. I should
       have accelerated the car to the right until it had enough velocity
      to reach the top of the left hill, then I should have accelerated the
       car to the left to reach the flag. In the next trial, I will
      accelerate the car to the right until it has enough velocity to reach
       the top of the left hill, then I will accelerate the car to the left
       to reach the flag.
Trial 1:
I will accelerate the car to the right until it has enough velocity to
    reach the top of the left hill, then I will accelerate the car to the
     left to reach the flag.
Trial 2:
In this environment, my plan was to accelerate the car to the right until
     it had enough velocity to reach the top of the left hill, then I
    should have accelerated the car to the left to reach the flag.
    However, I did not take into account the fact that the car may not
    have enough velocity to reach the top of the left hill. I should have
     monitored the car's velocity and adjusted my strategy accordingly.
    In the next trial, I will accelerate the car to the right until it
    has enough velocity to reach the top of the left hill, then I will
    monitor the car's velocity and adjust my strategy accordingly.

Answer:
Based on the current game state and memories of the task, the optimal
    action for the player to take would be to accelerate to the right (
    action 3). Therefore, the optimal action to take now is to push the
    cart to accelerate to the right (action 3).
```

Despite a few inaccuracies in the above few-shot example, Reflexion is still able to execute a superior strategy.

Table 10: Performance on different environments, deciders, and domain knowledge controlling levels. The first value in each cell is the median of different seeds and the value inside the parentheses represents the interquartile range while the last value is the maximum value.

| Environment | Decider Name | Lv1 | Lv2 | Lv3 | Lv4 | Lv5 |
|---|---|---|---|---|---|---|
| **Acrobot-v1** | EXE (Ours) | -200(0)/-200 | -200(0)/-200 | -200(0)/-200 | -200(0)/-200 | -200(0)/-200 |
| | CoT | -200(0)/-200 | -200(0)/-200 | -200(0)/-200 | -200(0)/-200 | -200(0)/-200 |
| | Naive-Actor | -200(0)/-200 | -200(0)/-200 | -200(0)/-200 | -200(0)/-200 | -200(0)/-200 |
| | Reflexion | -200(0)/-200 | -200(0)/-200 | -200(0)/-200 | -200(0)/-200 | -200(0)/-200 |
| | Self-Ask | -200(0)/-200 | -200(0)/-200 | -200(0)/-200 | -200(0)/-200 | -200(0)/-200 |
| | Self-Consistency | -200(0)/-200 | -200(0)/-200 | -200(0)/-200 | -200(0)/-200 | -200(0)/-200 |
| | SPP | -200(0)/-200 | -200(0)/-200 | -200(0)/-200 | -200(0)/-200 | -200(0)/-200 |
| | PPO | | | -96(14)/-70 | | |
| **Blackjack-v1** | EXE (Ours) | 15(1)/17 | 16(1)/17 | 16(1)/17 | 16(1)/17 | 15(0)/16 |
| | CoT | 14(2)/17 | 13(2)/16 | 14(1)/14 | 15(2)/16 | 11(0)/12 |
| | Naive-Actor | 18(0)/18 | 16(2)/18 | 16(1)/17 | 13(6)/18 | 10(0)/11 |
| | Reflexion | 13(0)/13 | 16(0)/16 | 16(1)/17 | 17(2)/17 | 11(0)/11 |
| | Self-Ask | 16(1)/17 | 9(4)/16 | 15(1)/15 | 13(2)/14 | 19(0)/19 |
| | Self-Consistency | 18(0)/19 | 15(4)/16 | 18(1)/18 | 15(2)/19 | 14(2)/15 |
| | SPP | 13(1)/13 | 11(3)/16 | 13(1)/14 | 12(0)/19 | 14(1)/16 |
| | PPO | | | 18(1)/20 | | |
| **CartPole-v0** | EXE (Ours) | 59(1)/131 | 62(28)/74 | 66(16)/92 | 52(39)/107 | 58(8)/67 |
| | CoT | 45(15)/53 | 24(16)/69 | 34(20)/54 | 48(14)/63 | 34(27)/69 |
| | Naive-Actor | 13(0)/13 | 9(0)/15 | 10(8)/18 | 10(0)/10 | 9(0)/9 |
| | Reflexion | 12(3)/21 | 9(0)/11 | 12(5)/46 | 12(6)/15 | 9(0)/9 |
| | Self-Ask | 28(1)/30 | 25(10)/40 | 35(6)/41 | 27(19)/65 | 69(18)/81 |
| | Self-Consistency | 41(7)/56 | 52(8)/73 | 38(3)/48 | 70(23)/72 | 30(15)/55 |
| | SPP | 53(14)/71 | 15(3)/22 | 17(17)/37 | 31(31)/76 | 51(26)/71 |
| | PPO | | | 200(0)/200 | | |
| **CliffWalking-v0** | EXE (Ours) | -1982(738)/-563 | -847(1363)/-354 | -127(110)/-118 | -2576(1889)/-26 | -347(-2675(297) |
| | CoT | -249(37)/-217 | -1685(990)/-200 | -2159(990)/-394 | -200(553)/-171 | -551(1040)/-23 |
| | Naive-Actor | -398(0)/-299 | -299(1089)/-200 | -398(198)/-200 | -20000(0)/-18515 | -200/-200(0) |
| | Reflexion | -893(297)/-497 | -5843(17325)/-695 | -19901(396)/-19505 | -200(0)/-200 | -794(0)/-596 |
| | Self-Ask | -299(0)/-186 | -497(87)/-299 | -314(115)/-200 | -398(99)/-299 | -337(548)/-126 |
| | Self-Consistency | -497(102)/-200 | -425(99)/-299 | -299(322)/-200 | -1808(145)/-28 | -134(110)/-15 |
| | SPP | -354(132)/-124 | -243(431)/-133 | -992(982)/-133 | -1883(693)/-319 | -344(154)/-51 |
| | PPO | | | -13/-13(0) | | |
| **LunarLander-v2** | EXE (Ours) | -931(279)/-696 | -926(93)/-743 | -735(113)/-656 | -825(72)/-755 | -713(143)/-685 |
| | CoT | -891(306)/-867 | -1003(0)/-1003 | -555(571)/-132 | -1033(0)/-1033 | -718(23)/-714 |
| | Naive-Actor | -804(0)/-804 | -1033(0)/-1033 | -1033(0)/-1033 | -1033(0)/-1033 | -1033(0)/-1033 |
| | Reflexion | -701(35)/-650 | -279(92)/-174 | -1033(0)/-978 | -248(40)/-107 | -1033(0)/-1033 |
| | Self-Ask | -452(59)/-274 | -889(432)/-795 | -1073(343)/-643 | -1376(87)/-1287 | -1457(6)/-1029 |
| | Self-Consistency | -1463(137)/-1026 | -1408(381)/-1007 | -550(549)/-483 | -1033(0)/-1033 | -747(25)/-726 |
| | SPP | -814(149)/-694 | -769(36)/-737 | -728(360)/-459 | -488(46)/-267 | -766(118)/-586 |
| | PPO | | | 262/261(103) | | |
| **MountainCar-v0** | EXE (Ours) | -200(0)/-120 | -200(0)/-139 | -200(76)/-116 | -200(70)/-123 | -200(0)/-135 |
| | CoT | -200(0)/-200 | -200(0)/-200 | -200(0)/-200 | -200(0)/-200 | -200(0)/-200 |
| | Naive-Actor | -200(0)/-200 | -200(0)/-200 | -200(0)/-200 | -200(0)/-200 | -200(0)/-200 |
| | Reflexion | -200(0)/-200 | -200(0)/-200 | -200(0)/-200 | -200(0)/-200 | -162(1)/-161 |
| | Self-Ask | -200(0)/-200 | -200(0)/-200 | -200(0)/-200 | -200(0)/-200 | -200(0)/-200 |
| | Self-Consistency | -200(0)/-200 | -200(0)/-200 | -200(0)/-200 | -200(0)/-200 | -200(0)/-200 |
| | SPP | -200(0)/-200 | -200(0)/-200 | -200(0)/-194 | -200(0)/-200 | -200(0)/-200 |
| | PPO | | | -90(1)/-87 | | |
| **MountainCarContinuous-v0** | EXE (Ours) | -13(1)/86 | -10(6)/-4 | 88(99)/91 | -8(1)/-6 | -12(3)/88 |
| | CoT | -15(1)/-14 | -11(0)/-10 | -6(5)/-2 | -9(3)/-7 | -20(0)/-20 |
| | Naive-Actor | -5(0)/-5 | -5(0)/-5 | -17(13)/-5 | -5(0)/-5 | -2(0)/-2 |
| | Reflexion | -5(0)/-4 | -3(1)/-2 | -7(11)/-4 | -5(0)/-5 | -20(0)/-20 |
| | Self-Ask | -12(1)/-11 | -12(1)/-11 | -15(3)/-10 | -12(4)/90 | -10(1)/-9 |
| | Self-Consistency | -14(1)/-14 | -10(3)/-9 | -13(1)/-12 | -15(1)/-14 | -20(0)/-20 |
| | SPP | -11(1)/-10 | -10(0)/-9 | -12(6)/-6 | -11(2)/-9 | -12(0)/-10 |
| | PPO | | | 0(0)/0 | | |
| **Taxi-v3** | EXE (Ours) | -821(0)/-605 | -227(27)/-218 | -227(630)/-200 | -236(459)/-218 | -740(216)/-605 |
| | CoT | -245(18)/-227 | -209(27)/-200 | -353(261)/-218 | -245(36)/-218 | -200(0)/-200 |
| | Naive-Actor | -1964(0)/-1964 | -200(0)/-200 | -200(0)/-200 | -200(0)/-200 | -200(0)/-200 |
| | Reflexion | -200(0)/-200 | -200(0)/-200 | -200(0)/-200 | -200(0)/-200 | -200(0)/-200 |
| | Self-Ask | -254(45)/-227 | -272(54)/-263 | -335(81)/-272 | -308(45)/-236 | -200(0)/-200 |
| | Self-Consistency | -200(18)/-200 | -200(0)/-200 | -290(72)/-200 | -200(9)/-200 | -200(0)/-200 |
| | SPP | -875(99)/-675 | -371(9)/-353 | -686(423)/-290 | -650(36)/-614 | -866(72)/-812 |
| | PPO | | | -144(12)/-140 | | |

## REFERENCES FOR SUPPLEMENTARY MATERIAL

Jacob Austin, Augustus Odena, Maxwell Nye, Maarten Bosma, Henryk Michalewski, David Dohan, Ellen Jiang, Carrie Cai, Michael Terry, Quoc Le, et al. Program synthesis with large language models. *arXiv preprint arXiv:2108.07732*, 2021.

Mark Chen, Jerry Tworek, Heewoo Jun, Qiming Yuan, Henrique Ponde de Oliveira Pinto, Jared Kaplan, Harri Edwards, Yuri Burda, Nicholas Joseph, Greg Brockman, et al. Evaluating large language models trained on code. *arXiv preprint arXiv:2107.03374*, 2021.

Marc-Alexandre Côté, Ákos Kádár, Xingdi Yuan, Ben Kybartas, Tavian Barnes, Emery Fine, James Moore, Ruo Yu Tao, Matthew Hausknecht, Layla El Asri, Mahmoud Adada, Wendy Tay, and Adam Trischler. Textworld: A learning environment for text-based games. *CoRR*, abs/1806.11532, 2018.

Sebastiaan De Peuter and Samuel Kaski. Zero-shot assistance in sequential decision problems. In *Proceedings of the AAAI Conference on Artificial Intelligence*, volume 37, pp. 11551–11559, 2023.

Luyu Gao, Aman Madaan, Shuyan Zhou, Uri Alon, Pengfei Liu, Yiming Yang, Jamie Callan, and Graham Neubig. Pal: Program-aided language models. In *International Conference on Machine Learning*, pp. 10764–10799. PMLR, 2023.

Ran Gong, Qiuyuan Huang, Xiaojian Ma, Hoi Vo, Zane Durante, Yusuke Noda, Zilong Zheng, Song-Chun Zhu, Demetri Terzopoulos, Li Fei-Fei, et al. Mindagent: Emergent gaming interaction. *arXiv preprint arXiv:2309.09971*, 2023.

Matthew Hausknecht, Prithviraj Ammanabrolu, Marc-Alexandre Côté, and Xingdi Yuan. Interactive fiction games: A colossal adventure. In *Proceedings of the AAAI Conference on Artificial Intelligence*, volume 34, pp. 7903–7910, 2020.

Dan Hendrycks, Steven Basart, Saurav Kadavath, Mantas Mazeika, Akul Arora, Ethan Guo, Collin Burns, Samir Puranik, Horace He, Dawn Song, and Jacob Steinhardt. Measuring coding challenge competence with apps. *NeurIPS*, 2021.

Wenlong Huang, Pieter Abbeel, Deepak Pathak, and Igor Mordatch. Language models as zero-shot planners: Extracting actionable knowledge for embodied agents. In *ICML*, 2022.

Takeshi Kojima, Shixiang Shane Gu, Machel Reid, Yutaka Matsuo, and Yusuke Iwasawa. Large language models are zero-shot reasoners. In *NeurIPS*, 2022.

Yujia Li, David Choi, Junyoung Chung, Nate Kushman, Julian Schrittwieser, Rémi Leblond, Tom Eccles, James Keeling, Felix Gimeno, Agustin Dal Lago, et al. Competition-level code generation with alphacode. *Science*, 378(6624):1092–1097, 2022.

Yuanzhi Liang, Linchao Zhu, and Yi Yang. Tachikuma: Understading complex interactions with multi-character and novel objects by large language models. *arXiv preprint arXiv:2307.12573*, 2023.

Erik Nijkamp, Bo Pang, Hiroaki Hayashi, Lifu Tu, Huan Wang, Yingbo Zhou, Silvio Savarese, and Caiming Xiong. Codegen: An open large language model for code with multi-turn program synthesis. In *The Eleventh International Conference on Learning Representations*, 2022.

Ofir Press, Muru Zhang, Sewon Min, Ludwig Schmidt, Noah A Smith, and Mike Lewis. Measuring and narrowing the compositionality gap in language models. *arXiv preprint arXiv:2210.03350*, 2022.

Noah Shinn, Federico Cassano, Beck Labash, Ashwin Gopinath, Karthik Narasimhan, and Shunyu Yao. Reflexion: Language agents with verbal reinforcement learning. *arXiv preprint arXiv:2303.11366*, 2023.

Mohit Shridhar, Xingdi Yuan, Marc-Alexandre Cote, Yonatan Bisk, Adam Trischler, and Matthew Hausknecht. {ALFW}orld: Aligning text and embodied environments for interactive learning. In *ICLR*, 2021.

Haotian Sun, Yuchen Zhuang, Lingkai Kong, Bo Dai, and Chao Zhang. Adaplanner: Adaptive planning from feedback with language models. *arXiv preprint arXiv:2305.16653*, 2023.

Guanzhi Wang, Yuqi Xie, Yunfan Jiang, Ajay Mandlekar, Chaowei Xiao, Yuke Zhu, Linxi Fan, and Anima Anandkumar. Voyager: An open-ended embodied agent with large language models. *arXiv preprint arXiv:2305.16291*, 2023a.

Xuezhi Wang, Jason Wei, Dale Schuurmans, Quoc Le, Ed Chi, Sharan Narang, Aakanksha Chowdhery, and Denny Zhou. Self-consistency improves chain of thought reasoning in language models. *arXiv preprint arXiv:2203.11171*, 2022.

Zhenhailong Wang, Shaoguang Mao, Wenshan Wu, Tao Ge, Furu Wei, and Heng Ji. Unleashing cognitive synergy in large language models: A task-solving agent through multi-persona self-collaboration. *arXiv preprint arXiv:2307.05300*, 2023b.

Zihao Wang, Shaofei Cai, Anji Liu, Xiaojian Ma, and Yitao Liang. Describe, explain, plan and select: Interactive planning with large language models enables open-world multi-task agents. *arXiv preprint arXiv:2302.01560*, 2023c.

Jason Wei, Xuezhi Wang, Dale Schuurmans, Maarten Bosma, Fei Xia, Ed Chi, Quoc V Le, Denny Zhou, et al. Chain-of-thought prompting elicits reasoning in large language models. In *NeurIPS*, 2022.

Frank F Xu, Uri Alon, Graham Neubig, and Vincent Josua Hellendoorn. A systematic evaluation of large language models of code. In *Proceedings of the 6th ACM SIGPLAN International Symposium on Machine Programming*, pp. 1–10, 2022.

Shunyu Yao, Howard Chen, John Yang, and Karthik Narasimhan. Webshop: Towards scalable real-world web interaction with grounded language agents. In *NeurIPS*, 2022.

Qinkai Zheng, Xiao Xia, Xu Zou, Yuxiao Dong, Shan Wang, Yufei Xue, Zihan Wang, Lei Shen, Andi Wang, Yang Li, et al. Codegeex: A pre-trained model for code generation with multilingual evaluations on humaneval-x. *arXiv preprint arXiv:2303.17568*, 2023.

Shuyan Zhou, Frank F Xu, Hao Zhu, Xuhui Zhou, Robert Lo, Abishek Sridhar, Xianyi Cheng, Yonatan Bisk, Daniel Fried, Uri Alon, et al. Webarena: A realistic web environment for building autonomous agents. *arXiv preprint arXiv:2307.13854*, 2023.

Xizhou Zhu, Yuntao Chen, Hao Tian, Chenxin Tao, Weijie Su, Chenyu Yang, Gao Huang, Bin Li, Lewei Lu, Xiaogang Wang, Yu Qiao, Zhaoxiang Zhang, and Jifeng Dai. Ghost in the minecraft: Generally capable agents for open-world environments via large language models with text-based knowledge and memory. *arXiv preprint arXiv:2305.17144*, 2023.

