# OpenReview forum: "Can Language Agents Approach the Performance of RL? An Empirical Study On OpenAI Gym"
_ICLR.cc/2024/Conference — Submitted to ICLR 2024_

### Official Review · Reviewer_tsqs · 2023-10-30

**Soundness:** 4 excellent
**Presentation:** 4 excellent
**Contribution:** 4 excellent
**Rating:** 8
**Confidence:** 3

**Summary:**

The manuscript introduces a novel benchmark, `TextGym`, which is based on the `OpenAI Gym` RL environments and can provide insightful assessment for the decision-making capabilities of language models.  The authors also compared various existing language agents based on `GPT-3`, and presented an `EXE` agent that strategically balances exploration and exploitation.

**Strengths:**

In general, this is a very well-organized and well-written paper, with solid contributions.

- The problem that the authors are investigating is a hot topic among LLM researchers, and people are in need of a unified benchmark to evaluate the decision-making capabilities of language models.
- The benchmark is novel, and has the potential to generate impact.
- The authors provided comprehensive and detailed empirical results.

**Weaknesses:**

- I would encourage the authors to avoid assertive and absolute claims, such as the first sentence "large language models lack grounding in external environments", and at the bottom of page 1, "we adopt `OpenAI Gym`, the most classic and prevalent...".  These arguments, though eye-catching, are debatable, and such debates are irrelevant to the central points that the authors are trying to make, so why not avoid them?
-  In the final paragraph on page 2, "after *sufficient* numerical experiments..."  I suppose that it should be left for the readers to decide whether the experiments are sufficient.
- Point 3) at the bottom of page 2 is confusing.  Which baseline do the language agents improve upon?  I suppose that the authors wanted to say that the performance can be improved *via* incorporating XXX, but it would be helpful to reorganize the sentence and clarify the point.
- The Problem Formulation section is somewhat redundant, as none of the math symbols are reused anywhere in the main text.

**Questions:**

- What is the main difference between the `EXE` language agent and other language agents, besides that the learner module provides exploration-exploitation balancing strategies in addition to how to exploit?  The reason that I am asking this question is that, if the actor and critic modules are identical to the other agents, the authors don't have to reiterate the descriptions on page 7.  The saved space can be dedicated to explaining how the learner module provides guidance on exploration and exploitation.
- It remains unclear to me (after reading the Appendix) how different levels of agents are implemented.  It would be useful to provide at least a brief description.

---

> ### Author Response · Authors · 2023-11-23
> **Response to reviewer tsqs**
>
> Thank you very much for recognizing our work, and your constructive comments and suggestions! We have revised our paper accordingly. Below, we will provide detailed responses to each point.
>
> > `Q1: I would encourage the authors to avoid assertive and absolute claims, such as the first sentence "large language models lack grounding in external environments", and at the bottom of page 1, "we adopt OpenAI Gym, the most classic and prevalent...". These arguments, though eye-catching, are debatable, and such debates are irrelevant to the central points that the authors are trying to make, so why not avoid them? In the final paragraph on page 2, "after sufficient numerical experiments..." I suppose that it should be left for the readers to decide whether the experiments are sufficient. Point 3) at the bottom of page 2 is confusing. Which baseline do the language agents improve upon? I suppose that the authors wanted to say that the performance can be improved via incorporating XXX, but it would be helpful to reorganize the sentence and clarify the point.`
>
> Thank you very much for your valuable comments! We have polished our text and avoided the over-claims.
>
> > `Q2: The Problem Formulation section is somewhat redundant, as none of the math symbols are reused anywhere in the main text.`
>
> Thanks for your advice! We have moved it to Appendix.
>
> > `Q3: What is the main difference between the EXE language agent and other language agents, besides that the learner module provides exploration-exploitation balancing strategies in addition to how to exploit? The reason that I am asking this question is that, if the actor and critic modules are identical to the other agents, the authors don't have to reiterate the descriptions on page 7. The saved space can be dedicated to explaining how the learner module provides guidance on exploration and exploitation.`
>
> Thanks you for your comments! We have added pseudocode of EXE and $5$ scenarios in our paper.
>
> > `Q4: It remains unclear to me (after reading the Appendix) how different levels of agents are implemented. It would be useful to provide at least a brief description.`
>
> We are apology for the missing description. We have added pseudocode for them in the revised version.

---

### Official Review · Reviewer_i4oJ · 2023-10-30

**Soundness:** 2 fair
**Presentation:** 1 poor
**Contribution:** 3 good
**Rating:** 3
**Confidence:** 3

**Summary:**

In this paper, the authors investigate whether LLMs can achieve the same level of performance as agents trained with RL in classical RL benchmarks. The authors consider various algorithms and settings for training Language agents in Gym environments. They introduce a pipeline for creating text-based versions of Gym environments, as well as 5 training scenarios. Finally, the authors propose their own algorithm, EXE, which improves on recent algorithms like Reflexion in solving their text-based versions of the Gym environments.

**Strengths:**

This paper asks an interesting and timely question. They introduce challenges around training Language Agents and discuss various levels of domain knowledge that can be provided to the agents. The proposed algorithm, EXE, can solve more tasks than the baselines, and comes close to PPO performance on some tasks, which is nice.

**Weaknesses:**

While the idea is timely, the methodology of the paper is unfortunately not very clear. The algorithm the authors propose is not introduced formally, but in charts. This makes it challenging to understand the algorithm, or why it works. Furthermore, the five training scenarios are not described in enough detail either. As a result, it’s not clear to me why L3 does the best out of the setups. It would also help if the authors motivate their algorithm better, as well as the reason why they expect it to work better than previous approaches.

It is also strange to me (if I understood this correctly) that the authors used GPT-4 to generate the python code that translates the Gym observations into text. This method seems error prone, and the authors do not demonstrate that this works, or why it’s even necessary in the first place.

Lastly, the experiments are not presented with enough detail. For instance:
* the number of seeds used to evaluate the agents is not reported.
* The learning curves or the converged performance of the PPO/SAC agents are not reported in the main text, but the Appendix.
* The threshold for saying that a task is solved is not defined in the main text either.


Minor points:
* The radar plot in figure 3B is very crowded.
* Parts of the paper are not so easy to follow. Consider simplifying language.

**Questions:**

* If SAC can solve tasks that PPO cannot solve in your setup, why are the PPO scores reported for the majority of the tasks instead of the SAC scores?
* I don’t see how some of these environments are partially observable (for instance Acrobot  and Cartpole) - this depends on how the states are translated into text. If they are indeed partially observable, it would be good to clarify how in the text.
* The authors say that Language agents are better in more intuitive and deterministic environments. Aren’t environments like Acrobot and LunarLandar deterministic (after the random first state)? And Blackjack, which is stochastic, can be solved quite well by EXE according to the Radar plot? What do the authors mean when they say an environment is intuitive?

While I like the idea of designing algorithms that make Language Agents better at solving RL tasks, I think the experiments and ideas of the paper need to be presented differently. The motivation for their algorithm is not clear, and the evaluation of the different agents does not seem very thorough. I therefore recommend that the paper is rejected as it currently is. I think the paper could be improved a lot by motivating the algorithm and a specific training scenario they recommend better. The evaluation of the standard RL algorithms like PPO and SAC could also be done more thoroughly. Finally, it would be great to understand why EXE works better than the Language Agents.

---

> ### Author Response · Authors · 2023-11-23
> **Response to reviewer i4oJ**
>
> Thank you very much for your constructive comments and suggestions. We have revised our paper accordingly. Below, we will provide detailed responses to each point.
>
> > `Q1: While the idea is timely, the methodology of the paper is unfortunately not very clear. The algorithm the authors propose is not introduced formally, but in charts. This makes it challenging to understand the algorithm, or why it works. Furthermore, the five training scenarios are not described in enough detail either. As a result, it’s not clear to me why L3 does the best out of the setups. It would also help if the authors motivate their algorithm better, as well as the reason why they expect it to work better than previous approaches.`
>
> Thank you very much for your valuable advices! We have improved our presentation of the algorithm and $5$ levels.
>
> > `Q2: It is also strange to me (if I understood this correctly) that the authors used GPT-4 to generate the python code that translates the Gym observations into text. This method seems error prone, and the authors do not demonstrate that this works, or why it’s even necessary in the first place.`
>
> Thanks for your helpful comments! We have moved it to the appendix.
>
> > `Q3: Lastly, the experiments are not presented with enough detail. For instance: the number of seeds used to evaluate the agents is not reported. The learning curves or the converged performance of the PPO/SAC agents are not reported in the main text, but the Appendix. The threshold for saying that a task is solved is not defined in the main text either.`
>
> Thank you very much for your questions!
>
> 1. we have extended our evaluations to $5$ seeds.
> 2. PPO and SAC performances are added.
> 3. The threshold description is moved to the main text.
>
> > `Q4: If SAC can solve tasks that PPO cannot solve in your setup, why are the PPO scores reported for the majority of the tasks instead of the SAC scores?`
>
> Thank you very much for your questions! SAC performances are added. There are also tasks that SAC can not solve.
>
> > `Q5: I don’t see how some of these environments are partially observable (for instance Acrobot and Cartpole) - this depends on how the states are translated into text. If they are indeed partially observable, it would be good to clarify how in the text.
> The authors say that Language agents are better in more intuitive and deterministic environments. Aren’t environments like Acrobot and LunarLandar deterministic (after the random first state)? And Blackjack, which is stochastic, can be solved quite well by EXE according to the Radar plot? What do the authors mean when they say an environment is intuitive?`
>
> This is a very interesting and important question! It is true that not all of the environments are partially observable. We have summarized the challenges for each environment in Appendix C.1 We have altered the informal description "non-intuitive" to "complex dynamic". For instance, it is easy to reason that the cart will move right if we push right when the cart speed is $0$. We call this dynamic simple. For `Acrobot`, it is hard to reason how the two links will behave when applying $-1$ torque to the actuated joint.
>
> > `Q6: I think the paper could be improved a lot by motivating the algorithm and a specific training scenario they recommend better. The evaluation of the standard RL algorithms like PPO and SAC could also be done more thoroughly. Finally, it would be great to understand why EXE works better than the Language Agents.`
>
> Thank you very much, it is a really good suggestion. In the revised version:
> 1. We take `CliffWalking` as our motivating example.
> 2. We make further analysis on EXE, especially comparing it to Reflexion.

---

### Official Review · Reviewer_9WGb · 2023-10-31

**Soundness:** 1 poor
**Presentation:** 1 poor
**Contribution:** 1 poor
**Rating:** 3
**Confidence:** 4

**Summary:**

The papers compares the performance of PPO and LLM-based AI agents on text-based version of Gym environments. The performance is evaluated by introducing different levels of domain knowledge into the LLM-based agent, and using a peculiar architecture (called EXE) for the LLM-based AI agent to interact with the environment.

**Strengths:**

Despite my generally negative judgement on the paper, it presents some positive features in terms of positioning and perspective. In particular, these are, in my opinion, the strengths of the paper:
- The study of how LLM-based AI agents should be evaluated, and how their performance can be compared to other classes of agents, is becoming an important research problem worth investigating.
- The principle of evaluating an LLM-based agent's performance when prompted with varying amounts of task-related information is sound and promising.

**Weaknesses:**

The paper presents many issues and limitations, both in terms of experimental protocols and rigor, and in terms of presentation. These are the main issues that I have identified:
- First and most importantly, the level of rigor in the empirical evaluation is far from being satisfactory. Performance comparisons are executed using a single seed only and not showing any form of confidence or error bars anywhere in the plots. A comparison of this kind is simply meaningless from the empirical standpoint, regardless of the funding constraints one might have to comply to, and invalidates all the claims and supposed findings contained in the paper. The sequential decision-making scientific community has made significant progress on evaluation rigor in the last few years, and I encourage the author to look at recent work (e.g., https://arxiv.org/abs/2304.01315 , https://arxiv.org/abs/2108.13264) and comply to the standards proposed there. The presence of LLMs or product-oriented APIs does not change the importance of rigorous scientific evaluation.
- Some details in the presentation are confusing or unnecessary. As a significant example, why should one highlight that GPT-4 generated the text version of the environment, if that one is not a feature that is being scientifically evaluated? As of now, it reads an attempt to yield to GPT-4 the responsibility concerning the correctness of the new environments that are being proposed. But this is totally irrelevant for the matter of the presentation of the scientific evaluation of a system: regardless of whether the code for evaluating the AI agents was generated by GPT-4 writing code, humans writing code, or copied from a random repository on the web, the only thing that matters is its correctness and alignment with the claims that the rest of the paper is trying to make.
- I believe a meaningful comparison of pretrained LLM-based systems with RL algorithms trained from scratch might not be as straightforward as the paper is trying to imply. LLMs are pretrained on all the internet, and the neural networks trained with RL start from scratch. It is still interesting to compare the performance of these different approaches, but saying that one approach is more sample efficient than another one does not do justice about their different nature and tradeoffs.

**Questions:**

- How are the claims in the paper statistically significant with a single seed and no error bars?
- Can you remove any mention to GPT-4 automatically generating environments?
- Why did you use PPO for some environments and SAC for some others?

---

> ### Author Response · Authors · 2023-11-23
> **Response to reviewer 9WGb**
>
> Thank you very much for your constructive comments and suggestions. We have revised our paper accordingly. Below, we will provide detailed responses to each point.
>
> > `Q1: First and most importantly, the level of rigor in the empirical evaluation is far from being satisfactory. Performance comparisons are executed using a single seed only and not showing any form of confidence or error bars anywhere in the plots. A comparison of this kind is simply meaningless from the empirical standpoint, regardless of the funding constraints one might have to comply to, and invalidates all the claims and supposed findings contained in the paper. The sequential decision-making scientific community has made significant progress on evaluation rigor in the last few years, and I encourage the author to look at recent work (e.g., https://arxiv.org/abs/2304.01315 , https://arxiv.org/abs/2108.13264) and comply to the standards proposed there. The presence of LLMs or product-oriented APIs does not change the importance of rigorous scientific evaluation.`
>
> Thank you very much for your comments. Please refer to **Response to all reviewers: Q1 and Q3.**
>
> > `Q2: Some details in the presentation are confusing or unnecessary. As a significant example, why should one highlight that GPT-4 generated the text version of the environment if that one is not a feature that is being scientifically evaluated? As of now, it reads an attempt to yield to GPT-4 the responsibility concerning the correctness of the new environments that are being proposed. But this is totally irrelevant for the matter of the presentation of the scientific evaluation of a system: regardless of whether the code for evaluating the AI agents was generated by GPT-4 writing code, humans writing code, or copied from a random repository on the web, the only thing that matters is its correctness and alignment with the claims that the rest of the paper is trying to make.`
>
> Thank you very much for your advices. We have moved it to the appendix and taken a more compact way to present it.
> However, we also want to clarify that the results that GPT-4 is able to generate a translator based on the documentation and template is very helpful for researchers who want to use language agents as alternatives to PPO.
>
> > `Q3: I believe a meaningful comparison of pretrained LLM-based systems with RL algorithms trained from scratch might not be as straightforward as the paper is trying to imply. LLMs are pretrained on all the internet, and the neural networks trained with RL start from scratch. It is still interesting to compare the performance of these different approaches, but saying that one approach is more sample efficient than another one does not do justice about their different nature and tradeoffs.`
>
> Thank you for your valuable suggestions! We have repositioned our paper as in **Response to all reviews: Q1** and motivated our comparisons as in **Response to all reviews: Q2**.
>
> > `Q4: Why did you use PPO for some environments and SAC for some others?`
>
> Thank you very much for your question! For `MountainCarContinuous`, PPO fails to generate expert-level performance trajectories due to the exploration challenge in `MountainCarContinuous`. PPO gets easily to stuck in sub-optimal strategy (not move) even with grid searching hyper-parameters. Although our main focus is the comparison of PPO, the infeasible expert trajectory will make the `Lv4` scenario infeasible. Thus we take SAC to generate the trajectories and we take it to construct the L4 scenario. See also in the **Response for reviewer DgbS.**

---

### Official Review · Reviewer_CAf4 · 2023-10-31

**Soundness:** 2 fair
**Presentation:** 3 good
**Contribution:** 2 fair
**Rating:** 5
**Confidence:** 3

**Summary:**

The authors create a TextGym benchmark conforming to the popular OpenAI Gym interface to evaluate LLM agents on some common RL environments, including CartPole, MountainCar variants, BlackJack and CliffWalking (see Fig. 3a for names of others).

The environments are converted into textual form to be able to be parsed by the language agents. This is done by using the official documentation of the environments and prompting GPT-4 with the info (the process is described in Fig. 5 in the Appendix).

The authors further design 5 scenarios with different amounts of prior knowledge that is provided to the language agents to be able to solve the environments - levels 1 through 5 with a higher number corresponding to more prior knowledge provided to the agents. Level 1 corresponding to zero prior knowledge, level 2 to providing non-optimal policies' trajectories to learn from (inspired by offline RL) - they use random policy rollouts for this scenario, level 3 to letting the agent interact with the environment for some time (inspired by RL) - they let the language agent interact with the environment for 5 episodes, level 4 to providing expert trajectories (inspired by imitation learning) - they use optimised RL agents from the Tianshou library for this scenario, level 5 to be provided expert human guidance in natural language to help the agent solve the environment.

They also come up with a unified architecture for such language agents with 3 components: actor, critic and learner (inspired by RL). They design a novel agent (the explore-exploit-guided language (EXE) agent) conforming to this architecture, with the learner and critic having long term memories of episodic transitions and the actor having a short term memory of the current episode's transitions. The learner guides the actor to explore or exploit based on criticism from the critic.

Experimentally, various language agents (Table 1) are benchmarked on their TextGym environments with performances normalized between -1 and 1. They show that the language agents can solve 5/8 environments and provide some inisghts into why and what agents and what scenarios performed better in the end.

**Strengths:**

The idea behind the benchmark and the various scenarios and unified architecture are well developed. The process of converting RL environments to a text interface and evaluating language agents on them is likely to lead to useful inisghts. I also find the different scenarios that have been designed intuitive. The paper is generally well presented  in terms of motivation and the problem tackled seems timely given the advent of LLMs.

**Weaknesses:**

I would say that lack of many seeds for a more thorough evaluation are the main weakness. The paper mentions in Appendix B.3 that only one seed is used for evaluation (except 100 for BlackJack). That feels too little given the kinds of environments in there do not seem expensive to me. Additionally, a few more environments could be evaluated given what seems to me to be not so compute-heavy experiments. Finally, the analyses in sections 5.2 and 5.3 seemed to lack in detail.

Minor:
OpenAI Gym is not an environment, it is library that contains an API for enviroments.

**Questions:**

Could the authors elaborate on the compute costs?

Given the fact that a lot of training is not needed for these experiments (as I understand it the authors use existing trained models for the LLM part and only train the RL agents with default hyperparameters in Tianshou, so no hyperparameter tuning is needed), I believe more seeds could be run and a wider range of environments could be included, a couple of Mujoco or Atari environments, for example. Running only one seed could also lead to noisier insights and make it harder to analyse the experiments. Could the authors clarify?

While I feel the paper is generally well written, the insights and analyses in sections 5.2 and 5.3 seem a bit high-level to me. For example in Fig. 3b: L3 agents actually did better than L5 and L4 agents, L4 was actually very bad even though it is supposed to be the second easiest scenario. L2 and L1 are sometimes better than L3 and L5 agents even though these are harder.  The reasons for such unintuitive results could be discussed in much more detail. The reason for lack of depth of analyses could be that a substantial portion of the paper is dedicated to presenting the benchmark and even Fig. 5 and the following code template in listing-1 are in Appendix even though in my opinion these are key to understanding the work. This makes me feel that maybe a journal would lend itself better to detailed presentations for a promising benchmark such as this. Another reason could be the lack of running many seeds which as mentioned above leads to noisier insights and analyses. Another example of a superfluous statement for me is the first key finding at the end of section 5.3. It seems to be a claim without proper evidence that EXE's exploration and exploitation capabilities helped it perform better. Could the authors say more on this?

In the problem formulation section 2.1, it is unclear to me what the index i is for?

The first case study in Appendix C was already hard for me to understand, so I did not try reading further here. I am not sure what agent was used here. Also, why was a part of the prompt hallucinated ("cliffs(Transversal interval from (3, 1) to (3, 10)")? It was provided by the authors I suppose and could not be hallucianted.

Could you provide a Code / Jupyter example for the work?

Why not show SAC in Fig. 3a for Taxi and MountainCar continuous environments as default PPO cannot solve them?

Maybe a table showing how many environments were solved by each agent in each scenario would be useful?

Could you also please explain "Specifically, we allocate 1 hour of effort to develop scenarios for a single agent in each environment, randomizing the sequence of language agents." in more detail?

---

> ### Author Response · Authors · 2023-11-23
> **Thanks for your detailed comments! Response to Reviewer CAf4**
>
> > `Q1: Compute and Time Costs`
>
> For detailed information on the compute and time costs, please refer to our **Response to all reviewers: Q4**.
>
> > `Q2: Number of Seeds and Range of Environments`
>
> We have used 5 seeds to make evauation of Language Agents in our revised manuscript. While hyperparameter tuning is not a concern for our experiments, the decision-making process with LLM-based agents is inherently resource-intensive. This restricts the scale of our experiments. For additional context, see **Response to all reviewers: Q3**.
>
> > `Q3: Depth of Analyses in Sections 5.2 and 5.3`
>
> We have streamlined the presentation for code generation and relocated the GPT-related content to the appendix to allow for a more concise main text. In Experiment section, we have added more analysis and many case studies are provided for each environment in the Appendix D.
>
> > `Q4: Problem Formulation Clarification`
>
> The issue with the index \( i \) in the problem formulation section will be clarified and we now move the formulation section to the appendix.
>
> > `Q5: Explanation of Scenario Development Effort`
>
> We have refined our explanation regarding the design of expert prompts for each environment and language agent. To minimize bias in comparative analysis, we randomized the sequence of language agent prompt design. This approach also addresses potential concerns about the fairness and adequacy of prompt design. However the expert guidance can hard to be controlled abosolutely fair and we can hard to predict how good it will be for a novel task. Thus our paper proposes `Lv2`, `Lv3`, and `Lv4` scenarios to control the domain knowledge and reduce human efforts.

---

### Official Review · Reviewer_DgbS · 2023-11-08

**Soundness:** 1 poor
**Presentation:** 1 poor
**Contribution:** 2 fair
**Rating:** 3
**Confidence:** 3

**Summary:**

The paper proposes a text interface that uses GPT-4 to translate eight OpenAI-Gym environments into text-based games, then tests a particular LLM, gpt-3.5-turbo0301, on the eight text-based games. Besides necessary verbal description of the current observation in the game, the LLM is also prompted with verbal suggestions/tips for game playing. The suggestion can be either hand-crafted by the authors (called Level-5 scenario in the paper), or from a text-based in-context learning method presented in the paper. In this method, we collect a few episodes of playing experience as input, then instruct a LLM to summarize and evaluate the playing experience as well as the decision policy behind it, and then instruct the LLM again to turn the summary/evaluation into a verbal suggestion for better game playing. The game-playing data used in this method can further be either from a random policy (called Level-2 scenario in the paper), or from self-playing in the RL manner (called Level-3 scenario), or from an expert policy obtained by PPO/SAC (called Level-4 scenario).

The paper calls the above method, EXE, if the LLM is instructed to make suggestion that encourages some exploration behavior in the game. The paper reports that, without the EXE trick, the GPT3.5-based language agents can barely solve any of the text-based games except for Blackjack-v1. With the EXE trick, the language agent achieves reasonable performance on 5 out of 8 environments under the Level-3 scenario, i.e. when the data for suggestion learning is collected in RL manner. Other suggestion-learning scenarios (L2,4,5) does not seem to work even with the EXE trick.

**Strengths:**

LLMs exhibit potential in general problem solving, as well as in some new learning forms such as learning from instruction and few-shot in-context learning. Utilizing LLMs in various problem domains beyond NLP is an interesting and hot topic in general. This paper reports some informative results to this end. It is especially intriguing to see that current LLMs can learn more effectively from experience data of its own, compared with learning from experience data or verbal instructions of better decision agents (corresponding to the superiority of L3 agents over L4 and L5 agents in Figure 4a).

**Weaknesses:**

**(a)** I think the general perspective of the paper is confusing. This is reflected in the paper title already: The authors ask if language agents is competitive to RL, but language-based agent and RL-based agent are not competing methods at all. In fact, the Lv3 language agent discussed in this paper -- which seem to be the only working language agents, according Fig. 4a -- is exactly a reinforcement learning procedure that improves decision policy from evaluative feedbacks collected from environmental interaction. You can't beat "RL" with a RL agent.

More importantly, as "an empirical study on OpenAI-Gym", the experiment includes no Mujoco or Atari task at all, which is a notable drawback as many representative tasks of OpenAI-Gym are in these two categories. Even for the chosen task categories, the tasks are still selected, with 4 out 12 missing in the paper. I am afraid the simple tasks considered in this paper cannot represent "the performance of RL" which is the target of the paper.

**(b)** Even if we re-position this paper as just a comparison between two specific methods (EXE vs PPO) on some specific simple tasks as a preliminary study, the current experiment setup may be still a bit biased. For example, I suspect the model size in PPO is orders-of-magnitude smaller than the GPT-3.5 in the language agents. Task-specific heuristics is provided (only) to the language agents. The chosen PPO implementation seems to have weird result on some tasks such as MountainCarContinuous. Results of some tasks in the same category are missing. Finally, when it comes to sample complexity, we should keep in mind that the LLMs are pretrained on huge data, thus it's not really a "5 vs 20K episodes" game; perhaps the paper can compare also with meta-RL agents such as Gato (Reed et al. 2022). See my Question 1~4 below for the detailed concerns.

**(c)** In terms of novelty, the paper may need to better contrast with prior works, especially with Reflexion (Shinn et al. 2023) which is similar to the proposed EXE method. Current appendix A.1 is not enough in this regard.

**(d)** The current presentation leaves many things unclear. See my Question 5~10 below for the detailed concerns here. Experimentation code is not released, which is not ideal as an "empirical study" that proposes a new "benchmark".

**Questions:**

1. In your experiment, what's the architecture of the policy model at the PPO side? How large is the model for PPO?

2. Did you try not prompting the LLM with the game description info and what's the performance? Although game description is indeed readily available for the particular tasks you considered, as you argued in Remark 1, the description is not necessarily available for other tasks in general. In the end, our goal is not to solve the particular tasks in Gym. We only use them as a benchmark to find method that hopefully works in the general case. If the game description is domain-specific knowledge crafted by human, prompting LLM with such task-specific descriptions is, in its essence, not much different from equipping LLM with gaming expertise from human. It thus feels a bit unfair to compare LLMs equipped with human-generated game description against RL agents that are autonomously learning fully on its own.

3. Why didn't you use SAC as the RL baseline for all the eight environments? Also, have you tried other PPO implementations on MoutainCarContinueous-v0?

4. Have you tried Pendulum (classic control), Frozen Lack (toy text), Bipedal Walker (Box 2D), and Car Racing (Box 2D), which are tasks in the same categories with the ones the paper studies? Since the TextGym code is generated automatically by GPT-4, and your experiment does not involve massive training, I guess encompassing these tasks should not be a huge burden?

5. How does the TexGym interface translate the termination condition? It seems the example code in Appendix B.1 does not process the termination information at all.

6. Is the actor-critic-learner framework in Section 3.3 used in all the 5 scenarios in Section 3.2? If so, how do the critic and learner work in L1 (no guidance) and L5 (expert guidance) scenarios which seem to involve no learning at all?

7. In L2, L3, and L4 scenarios, will the actor receives the 5-episode data as part of its prompt for action selection (in that case the actor will repeatedly receive the data in every gaming step even at testing time), or the 5-episode data is only used by the critic and the actor never sees the 5-episode data but only receives the decision suggestion extracted from that data?

8. What's the default learner? Appendix B.2 only gives the default critic.

9. In the first paragraph of Page 8, how are the scores between (0,1) mapped to the raw performance? A linear mapping? For reproducibility concern, please give the raw performance scores of the solvability threshold and sota thresholds used in your experiment.

10. Figure 3b and last paragraph of Page 8: from the picture it is not evident at all that "L3-Agents outperform their counterparts". All colored regions are stacked together and it's hard to tell. Please give data table for the exact performance scores.

11. Any explanation for why L4 result is worse than L3? Intuitively, the training data in L4 agents should be of higher quality (in terms of performance score) than those used in L3 agents. If the LLM can summarize useful principles from the latter, it should be able to do that in the former case too, intuitively.

12. In the paragraph below Figure 4, you said "for L1, L2, and L4 scenarios, EXE also outperforms other [language agents]". But to my current understanding, in these three scenarios the agent has no chance to explore the environment at all, isn't it? These three scenarios are pure offline scenarios, while the exploration-vs-exploitation trade-off becomes relevant mostly in online RL setting.

---

> ### Author Response · Authors · 2023-11-23
> **Thanks for your detailed comments! Response to Reviewer DgbS (1/4)**
>
> > `Q1: Scope of the Study`
>
> Thank you for your insightful suggestions. We have repositioned our paper to explore the potential of language agents as an alternative to PPO, which is reflected in the updated title: "Can Language Agents Be Alternatives to PPO? A Preliminary Empirical Study On OpenAI Gym". This revision also addresses the concern of "beat RL with RL".
> See details in Common Response Q1.
>
> > `Q2: Omission of Mujoco, Atari Tasks, and More`
>
> Please refer to **Response to all reviewers: Q2 and Q4**. The evaluation costs are significant and cannot be overlooked. We have also made attempts on FrozenLake, which is challenging due to its stochastic dynamics. Even with optimal actions, performance can be negatively impacted. Therefore, we have not pursued further evaluations in this area for now, considering the cost of evaluation.
>
> > `Q3: A Fairer Comparison Between Language Agents and PPO`
>
> We direct the reviewer to **Response to all reviewers: Q2** for a more detailed explanation.
>
> For details on PPO, please see Appendix C.4 in our revised manuscript. In our experiments, we utilized Tianshou in almost every environment and `Taxi-v3` with stable-baselines3. These were trained and evaluated in OpenAI Gym, not TextGym. Regarding model size, our policy model has 8,964 trainable parameters when there are 3 actions and the input dimension is 1. This figure is the sum of elements across all parameter tensors that require gradient computation.
>
> > `Q4: Additional Comparison with Reflexion`
>
> Firstly, we have provided pseudocode for both EXE and Reflexion in the Appendix to elucidate their differences and the novelty of EXE. There are three main distinctions:
> 1. EXE can generate suggestions without prior experience;
> 2. The critic in EXE is a language model that evaluates the trajectory based on suggestions guiding the actor's sampling process;
> 3. EXE specifically considers insights, exploration, exploitation, and their balance, whereas Reflexion focuses solely on score improvement.
>
> We also added case studies between them in Experiment section 4.4, with a more detailed analysis presented in the Appendix D.

---

> ### Author Response · Authors · 2023-11-23
> **Response to Reviewer DgbS (2/4)**
>
> > `Q5: Necessity of Game Descriptions`
>
> We've included additional scenarios in the Appendix D.2, where EXE's performance in environments such as Cartpole, MountainCar, and MountainCarContinuous was notably lower without the basic game description. This demonstrates that EXE's effectiveness may be constrained when essential game details are not provided.
>
> It's important to note that the game description is not crafted to enhance agent performance artificially; it typically exists as part of the task's framework. If a simulator for a task has been developed, it is reasonable to assume that a description of the game would be available. Human knowledge at Level 5 (L5) differs significantly from a mere game description; it is often constructed to intentionally boost the performance of Language Agents. For instance, in Cliffwalking, humans might outline the cliff locations, the goal, and even provide demonstrations for Language Agents—information that may not be readily available for a new task.
>
> > `Q6: Release of Code`
>
> Our code has been open-sourced.
>
> > `Q7: Clarity in Figure 3b and Supporting Data`
>
> We have revised Figure 3b for better visualization, offering a clearer representation of the performance metrics.

---

> ### Author Response · Authors · 2023-11-23
> **Response to Reviewer DgbS (3/4)**
>
> > `Q8: Explanation for L4 Results Being Worse Than L3`
>
> A8: The GPT model inherently exhibits a degree of stochasticity, which can affect the outcome. When it comes to expert data, a lack of diversity can impede the agents' ability to generalize, a phenomenon observed in contrasts between imitation learning and reinforcement learning. This is supported by our case studies (Appendix 4.4).
>
> > `Q9: Clarification on Scenarios L1, L2, and L4`
>
> A9: Indeed, L1, L2, and L4 are offline scenarios where agents do not actively explore the environment. In L1, EXE is unique in that it engages the learner, potentially aiding the actor with structured suggestions about exploration and exploitation. Supporting this, we observed in CliffWalking that EXE visited the goal in all five trials, compared to Reflexion, which did not reach the goal in any trial within the L1 scenario. For L2 and L4, the improvements in our updated version are modest and can be attributed to the structured suggestions provided by our model. In earlier versions, the notable performance gains in L2 and L4 were likely influenced by stochastic variation.
>
> > `Q10: Peculiar Performance of PPO in MountainCarContinuous`
>
> For MountainCarContinuous, we conducted a comprehensive grid search covering various hyperparameters:
>
> - Learning rate: `{1e-3, 1e-4, 1e-5}`
> - Discount factor: `{0.99, 0.95, 0.9}`
> - Weight for entropy loss: `{0.01, 0.05, 0.1}`
> - Number of repetitions for policy learning: `{10, 20}`
>
> Despite these efforts, the performance issues persisted. A significant challenge in this environment is the substantial penalty received by the agent when failing to reach the target, which often leads to a strategy of inactivity as a quick convergence solution. We also tested the implementation of PPO using stable baselines3.
>
> It is worth mentioning that our goal was not to fine-tune PPO to achieve superior performance, as extensive parameter searching can equate to expert guidance akin to the L5 scenario. Achieving excellent performance through extensive effort (akin to prompt designing) is possible but usually not the preferred approach.
>
> > `Q11: SAC Baseline and PPO Implementations on MountainCarContinuous-v0`
>
> We incorporated SAC results after searching for optimal alpha, learning rate, and gamma using the stablebaselines3 repository. While SAC was successful in `MountainCarContinuous`, it did not perform well in `CliffWalking`.
>
> For each game, we performed a grid search for:
>
> - Learning rate: `{1e-3, 1e-4, 1e-5}`
> - Discount factor: `{0.99, 0.95, 0.9}`
> - Entropy regularization coefficient: `{0.1, 0.2, 0.5}`
> - Auto-entropy regularization coefficient setting.
>
> Both PPO and SAC encountered challenges in certain tasks.
>
> _Table: SAC Performance Across Different Games_
>
> | Games | Performances |
> | --- | --- |
> | CartPole-v0 | `194.67 ± 56.89` |
> | LunarLander-v2 | `-7.68 ± 6.72` |
> | Acrobot-v1 | `-74.67 ± 0.22` |
> | MountainCar-v0 | `-89.33 ± 1.56` |
> | Blackjack-v1 | `13.33 ± 14.89` |
> | Taxi-v3 | `-200.00 ± 0.00` |
> | CliffWalking-v0 | `-200.00 ± 0.00` |
> | MountainCarContinuous-v0 | `93.23 ± 2.06` |
>
> ---

---

> ### Author Response · Authors · 2023-11-23
> **Response to Reviewer DgbS (4/4)**
>
> > `Q12: Termination Condition in TextGym Interface`
>
> `TextGym` utilizes a translator interface wrapped around OpenAI Gym to handle the termination conditions. The termination flag is managed by the gym simulator, and language agents are informed about the termination conditions within the game descriptions. This is particularly highlighted in Appendix B.1.
>
>
> > `Q13: Actor's Exposure to Episode Data in Scenarios Lv2, Lv3, and Lv4`
>
> In scenarios `Lv2`, `Lv3`, and `Lv4`, the actor does not receive the $5$-episode data for action selection; it only receives the suggestions derived from the data.
>
> > `Q14: Default Learner Specification`
>
> Details about the default learner are now specified in Appendix B.2, highlighted in blue for clarity.
>
> > `Q15: Mapping of Scores and Raw Performance Data`
>
> The raw performance scores are documented in Appendix D.1. The normalized scores used in the experiments are computed as described in the experiment section on page 7.

---

### Public Comment · ~Peter_Jansen1 · 2023-11-21
**Large existing body of literature on text games**

I thought I might offer the authors some pointers to the large existing body of literature at the intersection of reinforcement learning, large language models, and text-based virtual environments (and particularly those with OpenAI Gym interfaces) -- as (imho) it would improve the presentation of the paper, it's integration with existing formalisms/literature, and properly framing its contribution.

A large list of text game papers exists at textgames.org ( https://www.textgames.org/ ).  There are currently two survey papers of the field, that (in the context of this submission) help give a broad introduction of the simulators, environments, modeling formalisms (centrally POMPD), and common agent architectures.

In terms of simulators, nearly every text-based environment simulator mirrors the OpenAI Gym interface -- e.g. TextWorld, TextWorldExpress, Jericho (for the older interactive fiction games), ALFWorld, and ScienceWorld all do this.  Indeed most of the early text-game literature primarily made use of reinforcement learning models, before transitioning to hybrid RL + LLM models, and more recently transitioning simply to LLMs that are either zero-shot (e.g. ChatGPT), or fine-tuned using decision transformer or behavior cloning formalisms (essentially Markov models that include some number of steps of the observation/action/reward history, and conditioning on this to predict the next action that should be taken at a given step).  Given this submissions focus on Actor-Critic models, there are also a number of both recent and baseline actor-critic models for text-based gyms that would likely serve as relevant baselines, and frameworks to compare against in the context of the proposed model to help frame its contribution.  In the literature, you should be able to find a large number of comparisons of RL vs LLM models on text gym environments, either directly in some of the recent papers (e.g. ScienceWorld includes both RL and LLM baselines), or through a meta-analysis of performance across papers (some of which is already done in the systematic survey papers).  Similarly, in terms of the proposed method for generating the environments (i.e. supplying templated code snippets to GPT-4, and asking it to fill in the blanks) -- an extremely similar method was published in ByteSized32, at both a larger scale (i.e. number of games) and I believe (from my reading) scope (i.e. the ByteSized32 games are generating many hundreds of lines of Python).

While a very small amount of this literature is mentioned over two sentences in the Appendix in passing ("As large language models (LLMs) have demonstrated remarkable capabilities in generalization and planning, an array of executive benchmarks has been proposed to assess their proficiency. Initial benchmarks primarily focused on text-based games such as ALFWorld (Shridhar et al., 2021), Jericho (Hausknecht et al., 2020), and TextWorld (Cotˆ e et al. ´ , 2018)."), IMHO this doesn't do justice to the large body of existing literature that speaks to implementing text gyms in general, the relative performance differences between RL vs sequence2sequence/transformer models, the specific frameworks (e.g. actor-critic models) investigated in this submission, and the extremely similar use of code generation language models for generating text games based on references.

---

> ### Author Response · Authors · 2023-11-23
> **Response to the public comment**
>
> We genuinely appreciate your thoughtful feedback and guidance regarding the integration of our paper with existing literature and formalisms. Your provided insights will undoubtedly help us improve our overall presentation, better position our contributions, and enrich the related work section.
>
> We have taken your suggestions to heart, and in our revised version, we have included a more thorough discussion of TextGame within the main text as well as in Appendix B. By incorporating these references, we aim to provide a more comprehensive understanding of the existing literature at the intersection of reinforcement learning, large language models, and text-based virtual environments.
>
> We are grateful for your assistance in pinpointing the relevant literature in the field of text games, simulators, and actor-critic models. It has guided us to create a more robust understanding of the existing literature and has allowed us to better locate our work within that context.
>
> Once again, we express our sincere gratitude for your valuable input, which has greatly benefitted our paper. We look forward to any further remarks or suggestions you may have.
>
> -- Respect and Best Wishes from the authors :)

---

### Author Response · Authors · 2023-11-23
**Response to all reviewers (1/2)**

We thank all the reviwers for the insightful, constructive, and helpful reviews. We are pleased that reviewers found our paper to be:

1. well-written: "**well presented**" (CAf4), "**well-organized and well-written**" (tsqs);

2. contributing to the community: "**an important research problem worth investigating**" (9WGb), "**interesting and timely question**" (i4oJ), "**hot topic**", "**potential to generate impact**" (tsqs);

3. technically sound: "**reports some informative results**", "**intriguing to see that current LLMs can learn more effectively**" (DgbS), "**well developed**", "**lead to useful inisghts**" (CAf4), "**sound and promising**" (9WGb), "**which is nice**" (i4oJ), "**comprehensive and detailed empirical results**" (tsqs).

Here is a summary of major updates made to the revision:

1. **Re-positioned Scope**: In line with the constructive critiques, we have re-positioned our paper to investigate whether language agents can be alternative to PPO for sequential decision tasks. We have carefully revised our manuscript to temper our claims, ensuring they are substantiated and measured. (**Title, Abstract, and Section 1**)

2. **Expanded Experiments and Analyses**: We have enriched our experimental framework by incorporating additional seeds (totaling 5) to fortify the robustness of our results, now reporting both the median and the interquartile range. New case studies have been appended, and we have transparently presented the economic and temporal expenditures of our experiments. These enhancements have been meticulously documented. (**Section 4, Appendix D**)

3. **Enhanced Presentation**: The manuscript’s clarity and coherence have been augmented by:
    - Introducing pseudo code to elucidate scenarios and agent behaviors (**Section 2.2, Appendices C.2, C.3**),
    - Transferring ancillary content to the appendix to streamline the narrative (**Section 2.1**),
    - Focusing on a singular case study (CliffWalking) for a more potent introduction (**Sections 1 and 3**),
    - Refining figure aesthetics for improved visual comprehension (**Figure 3.b**),
    - Diligently avoiding any overstatements throughout the text.

Note the revision made is marked as $\textcolor{blue}{\text{blue}}$ in the revised version. Below, we will provide a unified response to the main concerns raised by the reviewers, which are listed as follows:

> `Q1: Clarification on the scope of the study.`

We have repositioned our paper to investigate **whether language agents can be alternatives to PPO**. Given the superior zero- and few-shot decision-making abilities of language agents, this question warrants critical examination. When a researcher is confronted with a decision-making problem, one potential approach is to reframe the problem as textual tasks and utilize language agents for resolution. To determine if this approach is a viable alternative to training PPO, we have selected environments from OpenAI Gym as our testbeds. Our paper contributes to this investigation with the **introduction of TextGym grounding, hierarchical control of domain knowledge, and the actor-critic-learner framework** for language agents. Additionally, we have developed a novel language agent named **EXE** to further enhance the performance of language agents. We conclude by comparing language agents in TextGym with PPO in OpenAI Gym and summarizing key findings from the empirical study. We believe this paper will **illuminate the potential for employing language agents in decision-making domains**.

Please note that our paper **does not** claim that language agents are superior to PPO. We are merely in the preliminary stages of exploring their potential in decision-making scenarios. The title and content have been updated to reflect this exploratory stage: "**Can Language Agents Be Alternatives to PPO? A Preliminary Empirical Study in OpenAI Gym**" For further discussion, we refer the reviewer to the revised Introduction section (highlighted in $\textcolor{blue}{\text{blue}}$).

---

> ### Author Response · Authors · 2023-11-23
> **Response to all reviewers (2/2)**
>
> > `Q2: Experiment motivations and fairness in comparisons.`
>
> In considering alternatives to PPO, we posit the question: "**Should language agents be regarded as alternatives to PPO for sequential decision tasks when some foundational knowledge is present?**" Our aim is **not to assert the superiority** of language agents but rather to explore their potential applicability in this field. We proceed with the assumption that a researcher has ready access to a simulator and its accompanying documentation, and that the model size and the pre-training data of the LLM are not primary concerns when selecting an approach.
> Although "Gato"-like meta-RL methods may appear akin to language agents, their **lack of open-source availability** and their relative novelty in addressing new sequential decision tasks make them less prevalent in current research practices.
>
> The prospect of a fair comparison between language agents and PPO is indeed compelling. Yet, the marked disparities between the two present challenges. For instance, it is **impractical to align PPO and language agents in terms of network architecture or model size** due to their distinct learning preferences. Consequently, our comparisons are made from the standpoint of practical users, who might weigh different factors when choosing between these methods.
>
> > `Q3: Rigor and reproducible numerical results.`
>
> Our experiment now incorporates numerical results from **$5$ different seeds** to improve reliability, along with the interquartile range. We have **open-sourced the code** in this anonymous repository: https://anonymous.4open.science/r/Text-Gym-Agents-5220. Additionally, we have **provided detailed descriptions of our implementations** in the appendices, ensuring the reproducibility of our work.
>
> **Task selection from OpenAI-Gym**: We do not involved specific tasks for our experiments for two main reasons:
> 1. **Cost of Evaluation**: The evaluation process is resource-intensive, both financially and temporally. For example, we excluded the `BipedalWalker` environment from our assessment due to its requirement of 2,000 steps per episode, which incurs a cost **ten times greater** than other tasks.
> 2. **Complexity of Language Grounding**: Certain tasks present substantial language grounding challenges. In environments where the difficulty is pronounced, such as `FrozenLake`, we limit our reporting to partial results without repeated experiments to maintain feasibility. Additionally, tasks with image-based states, like many `Atari` games or `CarRacing` in `Box2D`, require a distinct approach to language grounding. We recognize these as complex issues and have earmarked them for future investigation.
>
> As noted in the **Response to all reviewers: Q4**, these constraints of cost and complexity have shaped our experimental design, focusing our resources on the most informative and manageable tasks.
>
> > `Q4: Time and economic costs of the experiments.`
>
> The significant time and economic investments our experiments required are noteworthy, emphasizing the value of the preliminary results we have achieved and the rationale for the scope of our study. We anticipate that these initial findings will spur further exploration in the field. Constraints due to OpenAI API's rate limits resulted in a total inference time of approximately **64 hours** for our experiments with GPT-3.5. The financial costs are also considerable; as indicated in the table provided, the expenses for conducting these experiments on GPT-3.5 amounted to roughly **2,611 dollars**. This data underscores the extensive resources necessary for such research and highlights the importance of our foundational work. Following table show the **toal costs on time and economic for each environment** and **total time spent on each language agent**, respectively.
>
> | Game                     | Time Spend(s) | Economic Cost ($) |
> |--------------------------|---------------|-------------------|
> | Acrobot-v1               | 49,830         | 565               |
> | Blackjack-v1             | 179           | 49                |
> | CartPole-v0              | 8,000          | 74                |
> | CliffWalking-v0          | 23,616         | 258               |
> | LunarLander-v2           | 38,309         | 415               |
> | MountainCar-v0           | 35,979         | 371               |
> | MountainCarContinuous-v0 | 41,571         | 390               |
> | Taxi-v3                  | 36,072         | 489               |
> | Total                    | 233,556        | 2,611              |
>
> | Decider          | Time Spend(s) |
> |------------------|---------------|
> | CoT              | 18,115         |
> | EXE (Ours)       | 12,161         |
> | Naive-Actor      | 9,291          |
> | Reflexion        | 12,387         |
> | SPP              | 63,095         |
> | Self-Ask         | 44,640         |
> | Self-Consistency | 64,743         |
> | Total            | 224,432        |

---

### Meta-Review · Area_Chair_t2Hb · 2023-12-07

**Metareview:**

This paper compares the relative abilities of PPO agents and a novel LLM-based agent (EXE) on a set of established benchmark reinforcement learning tasks. Whilst the topic is timely and likely of broad interest to the ICLR community, many issues raised by reviewers were not sufficiently addressed in time for careful review. The paper has already continued to improve and we encourage the authors to continue pursuing the topic, addressing the feedback given and considering re-submission to a future conference.

**Justification For Why Not Higher Score:**

+ Too many unresolved issues raised by reviewers suggesting promising, early work not yet ready for formal publication

**Justification For Why Not Lower Score:**

N/A

---

### Decision · Program_Chairs · 2024-01-16

Reject